# Polar coupling enabled nonlinear optical filtering at MoS$_2$/ferroelectric heterointerfaces

Dawei Li [1,4], Xi Huang[2,4], Zhiyong Xiao [1], Hanying Chen [1], Le Zhang[1], Yifei Hao[1], Jingfeng Song[1], Ding-Fu Shao[1], Evgeny Y. Tsymbal [1,3], Yongfeng Lu [2,3✉] & Xia Hong [1,3✉]

Complex oxide heterointerfaces and van der Waals heterostructures present two versatile but intrinsically different platforms for exploring emergent quantum phenomena and designing new functionalities. The rich opportunity offered by the synergy between these two classes of materials, however, is yet to be charted. Here, we report an unconventional nonlinear optical filtering effect resulting from the interfacial polar alignment between monolayer MoS$_2$ and a neighboring ferroelectric oxide thin film. The second harmonic generation response at the heterointerface is either substantially enhanced or almost entirely quenched by an underlying ferroelectric domain wall depending on its chirality, and can be further tailored by the polar domains. Unlike the extensively studied coupling mechanisms driven by charge, spin, and lattice, the interfacial tailoring effect is solely mediated by the polar symmetry, as well explained via our density functional theory calculations, pointing to a new material strategy for the functional design of nanoscale reconfigurable optical applications.

[1] Department of Physics and Astronomy, University of Nebraska-Lincoln, Lincoln, NE 68588-0299, USA. [2] Department of Electrical and Computer Engineering, University of Nebraska-Lincoln, Lincoln, NE 68588-0511, USA. [3] Nebraska Center for Materials and Nanoscience, University of Nebraska-Lincoln, Lincoln, NE 68588-0298, USA. [4]These authors contributed equally: Dawei Li, Xi Huang. ✉email: ylu2@unl.edu; xia.hong@unl.edu

The heterointerface between two functional materials presents a powerhouse of various emergent quantum phenomena and novel functionalities. Two notable examples are the complex oxide epitaxial interfaces[1] and van der Waals (vdW) heterostructures[2], with the former hosting interfacial magneto-electric coupling[3], gate-tunable two-dimensional (2D) super-conductivity[4] and topological states[5], and polar vortices[6,7], and the latter leading to the discoveries of the long sought after Hofstadter butterfly[8-10], moiré excitons[11-13], and correlation-driven quantum phase transitions[14,15]. An even broader spectrum of functional properties can emerge at the heterointerface between these two distinct materials, a territory yet to be fully explored. Like the ferroelectric oxides, monolayer (1L) transition metal dichalcogenides (TMDCs) such as $MoS_2$ are noncentrosymmetric and possess polar axes. The associated functional phenomena, including piezoelectricity[16] and polar metal switching[17], have drawn substantial research interests recently. When 2D TMDC is interfaced with a ferroelectric gate, the spontaneous ferroelectric polarization offers the unique opportunity to induce nonvolatile charge modulation in the channel[18-20]. Combining the polarization doping with nanoscale ferroelectric domain patterning further allows local tuning of the electronic[20-22] and optical properties[23-25] of the vdW channel layer. Beyond the charge-mediated interfacial coupling, synergy between the polar nature of TMDCs and ferroelectric has never been explored to date.

In this work, we report an unconventional nonlinear optical filtering effect enabled by the polar symmetry of 1L $MoS_2$ and a neighboring ferroelectric $PbZr_{0.2}Ti_{0.8}O_3$ (PZT) thin film. The heterostructure exhibits either strong enhancement or substantial quenching of the reflected second-harmonic generation (SHG)

response at the ferroelectric domain walls (DWs), which reveals the intricate coupling of the polar axis of $MoS_2$ with the chiral rotation of the surface dipole at the DWs, as modeled via our density functional theory (DFT) calculations. Unlike the exten-sively studied interfacial coupling mechanisms driven by charge, spin, and lattice[18], this tailored SHG signal is solely mediated by symmetry, pointing to a widely applicable strategy for achieving designate optical functionalities in noncentrosymmetric materials.

## Results

**Characterization of 1L $MoS_2$/PZT heterostructures.** Figure 1a shows the experimental set-up for the SHG imaging of the $MoS_2$–ferroelectric heterostructure. 1L $MoS_2$ flakes were mechanically exfoliated from bulk crystals on Gel-Films and identified via the frequency difference $\Delta$ between the $E_{2g}^1$ and $A_{1g}$ modes in the Raman spectrum (Fig. 1b, Supplementary Fig. 6b). The crystalline orientation of $MoS_2$ was identified on the Gel-Film by polarized SHG measurements (Supplementary Fig. 4b). For the ferroelectric layer, we worked with 20–50-nm-thick epi-taxial PZT thin films deposited on (001) $SrTiO_3$ substrates, with $La_{0.67}Sr_{0.33}MnO_3$ (LSMO) (10 nm) buffer layers serving as the bottom electrode ("Methods"). The PZT films are (001)-oriented with out-of-plane polar axis (Supplementary Fig. 1). Selected 1L $MoS_2$ flakes were transferred on top of the PZT film above a region patterned with a series of square domains with alternating up ($P_{up}$ or [001]) and down ($P_{down}$ or [00$\bar{1}$]) polarization. Fig-ure 1c shows the piezoresponse force microscopy (PFM) phase image of the domain pattern on a 50-nm PZT before the $MoS_2$

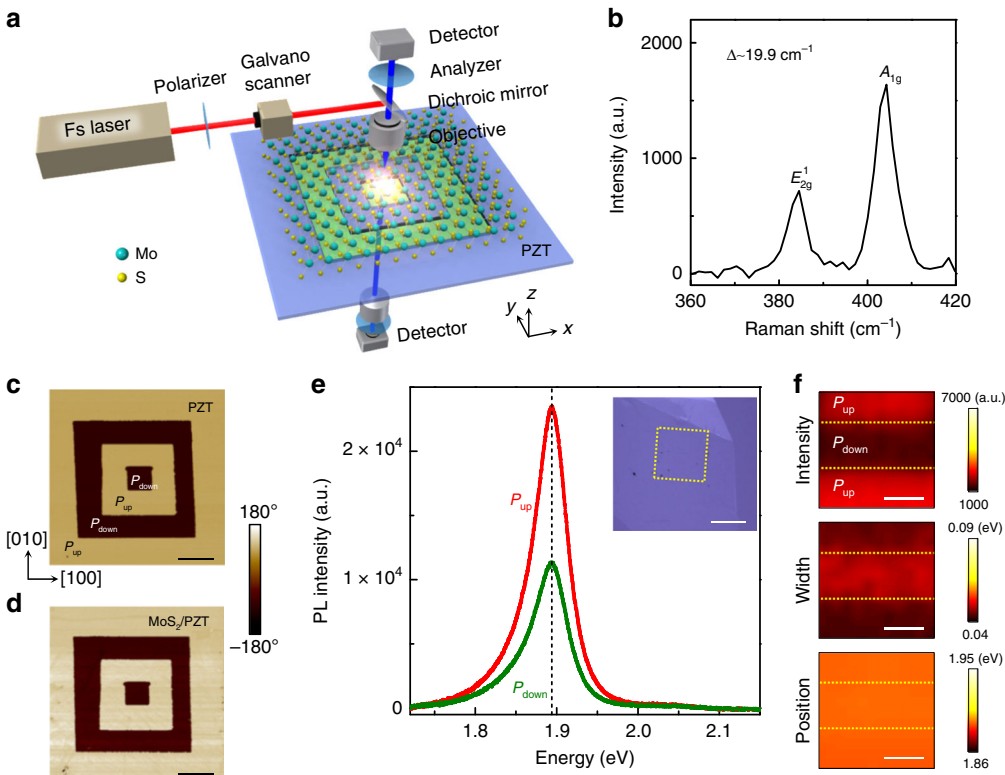

**Fig. 1 Characterization of 1L $MoS_2$/PZT heterostructures. a** Schematic of the SHG experimental set-up. The laboratory coordinate system is shown as inset. **b** Raman spectrum of a 1L $MoS_2$ flake on gel film showing $E_{2g}^1$ mode at 384.0 cm$^{-1}$ and $A_{1g}$ mode at 403.9 cm$^{-1}$. **c, d** PFM phase images of **c** square domains written on a PZT film and **d** the same region with a 1L $MoS_2$ transferred on top. Inset: Crystalline orientation of PZT. The scale bars are 3 μm. **e** Room temperature PL spectra of the 1L $MoS_2$ on the $P_{up}$ and $P_{down}$ domains shown in **d**. The domain region is outlined in the optical image of the sample (inset). The scale bar is 10 μm. **f** PL mapping of the peak intensity (upper), width (middle), and position (lower) on a 1L $MoS_2$/PZT sample in a region with both $P_{up}$ and $P_{down}$ domains. The dotted lines mark the DW positions. The scale bars are 2 μm.

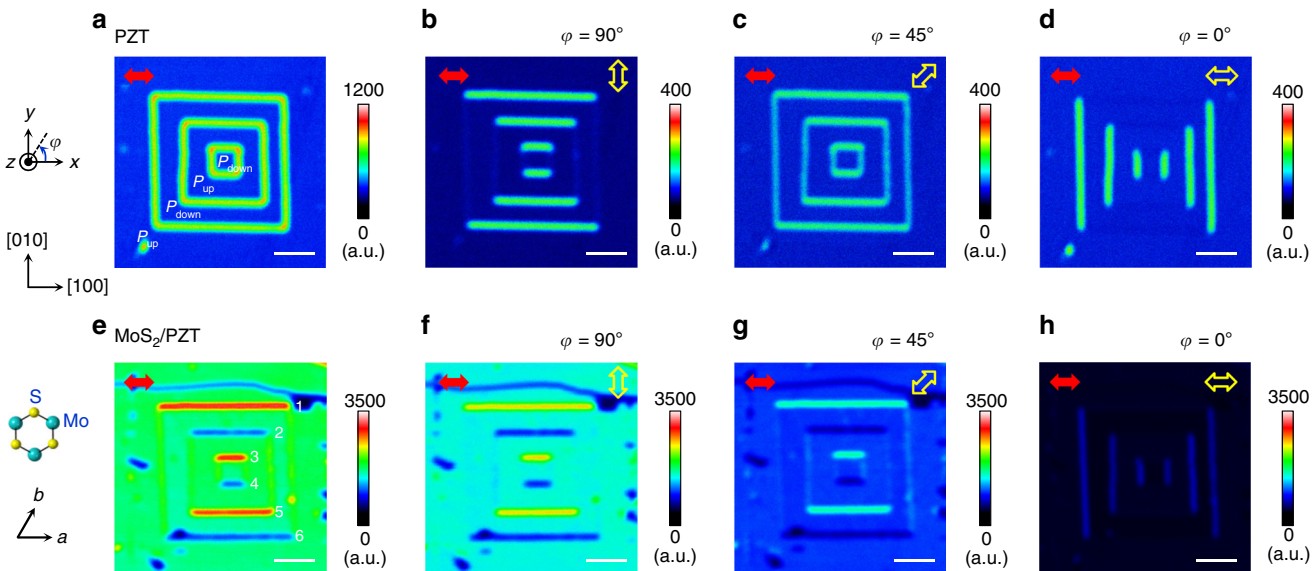

**Fig. 2 Reflected SHG response of domains on PZT with and without MoS₂ top layer. a–d** SHG mapping of the domain structure shown in Fig. 1c taken **a** with no analyzer applied and **b–d** with an analyzer applied at different angles $\varphi$ (yellow open arrows) with respect to the incident light polarization (red solid arrows). The excitation laser power is 30 mW. **e–h** SHG mapping of the same domain structure with a 1L MoS₂ flake transferred on top taken with the same polarizer and analyzer settings as in **a–d**, respectively. The excitation laser power is 20 mW. The scale bars are 3 μm. The crystalline orientations of PZT and MoS₂ are shown as insets.

transfer, where the horizontal (vertical) DWs are along the [100] ([010]) orientation of PZT. During transfer, the $a$-axis of MoS₂ (zigzag orientation) was aligned with the horizontal DWs ([100] orientation of PZT) (see "Methods" for transfer details). As shown in Fig. 1d, the presence of the MoS₂ top layer does not alter the underneath domain structure. This is not surprising as the PZT film is exposed to the ambient condition prior to the transfer, where the polarized surface-bound charge can be well screened by charged adsorbates[26,27]. The MoS₂ flake is deposited on top of the domain structure via a dry-transfer approach, which should not affect this surface screening layer on PZT.

Figure 1e compares the photoluminescence (PL) spectra of MoS₂ obtained from the regions on the $P_{up}$ and $P_{down}$ domains, with the corresponding PL mapping shown in Fig. 1f. While there is no change in the peak position, both the PL intensity and width exhibit strong dependence on the PZT polarization state. The region above the $P_{up}$ domain exhibits higher PL intensity, narrower peak width, and a reduced ratio between the trion and neutral exciton populations (Supplementary Fig. 5). Such modulation of PL spectra in TMDCs via neighboring ferroelectric domains has previously been attributed to the polarization-induced doping effect[23,24] and confirms the close interfacial contact between MoS₂ and PZT in our samples. The relative strength of the modulation, however, can be affected by the interfacial charge screening condition for PZT[27] and thus depends on the preparation details of the composite structures (Supplementary Note 2)[26].

1L MoS₂ exhibits strong nonlinear optical responses, such as SHG[21,28–30] and sum-frequency generation[21,30], due to the lack of inversion symmetry. For normal incident light (800 nm center wavelength), we observed strong SHG response (~400 nm) from the 1L MoS₂ flakes on Gel-Films, which conforms to the rotational symmetry of the lattice (Supplementary Fig. 4b–d). For the PZT films, as the incident light is a transversely polarized (within $x–y$ plane) electromagnetic wave propagating along the polar axis ($-z$-direction or $[00\bar{1}]$ orientation of PZT), there is no SHG response on the uniformly polarized domains. As shown in the SHG mapping image

(Fig. 2a), prominent SHG signals have only been observed at the DWs, consistent with previous reports on PZT thin films[31,32], which suggests the existence of an in-plane polarization ($p_{||}$) facilitated by the DW. The width of the detected SHG signal is about 300–400 nm, which approaches the diffraction limit at this wavelength and the resolution of the SHG microscope ("Methods"). To determine the orientation of $p_{||}$, we performed SHG imaging with an analyzer applied at various orientations, i.e., making angle $\varphi = 90°$, 45°, and 0° with respect to the incident light polarization ($x$-axis). As shown in Fig. 2b–d, the SHG response can only be detected when the analyzer can be projected along the direction perpendicular to the DW. This means that $p_{||}$ is residing in a plane normal to the DW, similar to the Néel-type chiral DW[31]. In bulk PZT, the 180° DWs are known to be at the unit cell scale[33,34], and such chiral DW is not energetically favorable. Continuous rotation of local dipoles, however, can be stabilized at the surface of PZT thin films by depolarization field[35], resulting in a net lateral polarization. For both MoS₂ on Gel-Films and bare PZT, the SHG signals detected in the transmission mode exhibit qualitatively similar behavior as in the reflection mode (Supplementary Figs. 3 and 4c, d).

**Reflected SHG response of 1L MoS₂/PZT heterostructures**. We then mapped the SHG response of the 1L MoS₂/PZT heterostructure. Figure 2e shows the reflected SHG mapping taken on the same domain structure in PZT with the 1L MoS₂ transferred on top. The imaging condition is similar to that used in Fig. 2a, i.e., with incident light polarization along $x$-axis ($a$-axis of MoS₂) and no analyzer applied. As expected, we observed strong SHG intensity from MoS₂ on the uniformly polarized $P_{up}$ and $P_{down}$ domains. Unlike the PL data (Fig. 1e, f), no prominent difference in the SHG signal has been observed in the regions on the $P_{up}$ and $P_{down}$ domains, confirming that the signal is not affected by the interfacial charge coupling between MoS₂ and PZT. At the DWs, however, the heterointerface produces a filtering effect for the reflected SHG that not only selects the light polarization, similar

to that of a vertical analyzer (Fig. 2b), but also the DW chirality. Along the vertical ([010]) DWs, the SHG signal is at a similar level to those on the $P_{up}$ and $P_{down}$ domains. This is in sharp contrast to those observed on bare PZT, where the vertical DWs have similar intensity as those from the horizontal ([100]) DWs (Fig. 2a). The horizontal DWs, more interestingly, exhibit alternating enhancement and suppression of the SHG signals. At the set of DWs labeled as 1, 3, and 5, the SHG response is about two times of those on the $P_{up}$ and $P_{down}$ domains. At the other set of DWs (labeled as 2, 4, and 6), the SHG response is substantially quenched. In Fig. 2e, the MoS$_2$ flake shows several cracked regions resulting from the transfer, exposing the bare PZT underneath. The fact that the SHG intensity at the even-numbered DWs is comparable to these regions indicates that the emission from MoS$_2$ is close to be entirely canceled by the presence of these DWs. The tailoring of the reflected SHG signal at the DW is a robust effect and has been observed in multiple 1L MoS$_2$ samples. Similar tuning pattern is also observed on three- and five-layer MoS$_2$ flakes on PZT and is absent in bilayer and four-layer MoS$_2$ (Supplementary Fig. 6), which reveals the essential role of the noncentrosymmetric symmetry of MoS$_2$ in the observed effect. In samples with odd-layer MoS$_2$, the modulation strength decreases with increasing layer number, consistent with fact that the SHG signal of MoS$_2$ attenuates rapidly in thicker films[29].

Figure 2f–h show the SHG mapping with an analyzer applied at the same angles $\varphi$ as in Fig. 2b–d, respectively. At $\varphi = 90°$ (Fig. 2f), the image shows qualitatively similar SHG behaviors as in Fig. 2e, confirming that the signals at the DWs are linearly polarized, with the polarization perpendicular to the DW. At $\varphi = 45°$, even though the intensity of the SHG signal is significantly suppressed for both the domain and DW regions, the relative relation between them remains the same (Fig. 2g). Only when the SHG signal of MoS$_2$ is fully quenched by a parallel analyzer at $\varphi = 0°$ (Supplementary Fig. 4b) does the signal from the vertical DWs of PZT become appreciable (Fig. 2h).

The alternately enhanced or suppressed SHG signals can be well correlated to the in-plane polarization of the DWs. A clear difference between the odd- and even-numbered DWs is the arrangement of the domains that they separate. The odd DWs are accompanied with top $P_{up}$ and bottom $P_{down}$ domains, opposite to the distribution for the even DWs. To conform to the bulk polarization change, the surface polarization at the vicinities of the odd and even DWs is expected to have opposite chirality (Fig. 3a), with the corresponding $\vec{p}_\parallel$ pointing to $-y$ and $+y$ directions, respectively. The orientation of $\vec{p}_\parallel$ itself does not have an impact on the intensity of the SHG response ($I \propto t|\vec{E}|^2$), as clearly shown for bare PZT in Fig. 2a. The presence of a 1L MoS$_2$ on top, however, modifies the polar symmetry of the heterointerface. One of the polar, armchair directions of the MoS$_2$ flake is along the $y$-axis, which is either parallel or anti-parallel to $\vec{p}_\parallel$ for the horizontal ([100]) DWs, depending on the clarity. The enhanced or suppressed SHG response can thus be attributed to the alignment of the polar axis of MoS$_2$ ($\vec{P}_{MoS_2}$) with the in-plane polarization at the PZT DWs ($\vec{P}_{DW}$).

Next, we compared the SHG response of 1L MoS$_2$ interfaced with 20, 30, and 50 nm PZT films (Supplementary Fig. 7). Despite the different PZT thicknesses, all heterostructures exhibit qualitatively similar SHG responses, with alternating enhancement and suppression of the SHG signal observed at the horizontal ([100]) DWs and unappreciable SHG contrast observed at the vertical ([010]) DWs. This result further confirms the interfacial nature of the DW's tailoring effect. In fact, the 180° DW in bulk PZT is on the order of a couple of unit cells and does

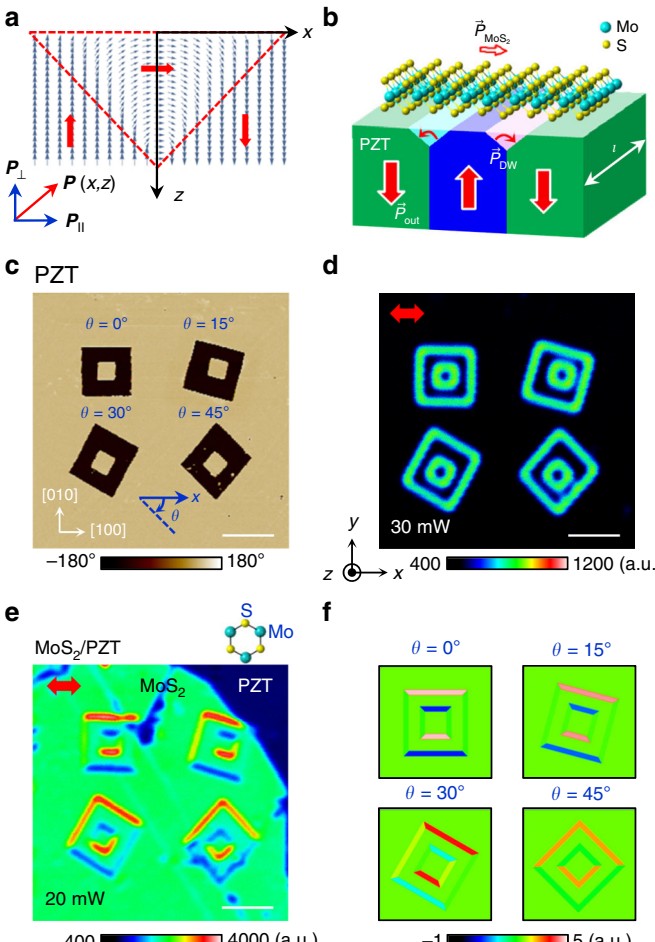

**Fig. 3 Effect of polar alignment on the reflected SHG response. a**, **b** Schematics of **a** a flux-closure domain (boundary outlined by red dashed lines) at ferroelectric surface above a 180° DW and **b** the polar alignment at the 1L MoS$_2$/PZT interface. The arrows mark the local polarization orientation. **c** PFM phase image of square domains written on PZT along different angle $\theta$, with the corresponding SHG images taken **d** before and **e** after a 1L MoS$_2$ transferred on top. The excitation laser power is labeled on the plot. The red arrows mark the incident light polarization. There is no analyzer applied. The scale bars are 3 μm. **f** Simulated SHG amplitude at 1L MoS$_2$/PZT interface for the same domain structures in **c**–**e**. The crystalline orientations of PZT and MoS$_2$ and the laboratory coordinate system are shown as insets.

not acquire an in-plane component[33,34]. The chiral rotation of the local dipole, which is critical for forming the in-plane polarization, can only be stabilized at the surfaces/interfaces (Fig. 3a) due to the presence of strong depolarization field[35,36]. For example, previous transmission electron microscopy (TEM) studies have revealed a flux-closure polar structure at the surface of the DW in PZT thin films[35] and even emergence of polar vortices in PbTiO$_3$/SrTiO$_3$ superlattices[6,7], where theoretical modeling has pointed to the dominant role of the interface contribution to the polar anomaly[7].

**Theoretical modeling of interfacial polar coupling.** To examine the feasibility of the interfacial polar coupling scenario, we exploited a phenomenological model to estimate the net in-plane polarization at the DW based on the TEM result[35], considering a triangle-shaped flux-closure domain structure that hosts continuous electric dipole rotations at the surface of a 180° DW

(Fig. 3a). Compared with an *a*-domain-like DW configuration induced by the local electric field of a biased atomic force microscope (AFM) tip (Supplementary Fig. 10), this model depicts a dipole distribution with comparable (if not smaller) spatial extension but lower electrostatic energy[34]. The bulk values of the out-of-plane ($P_{out}$) and in-plane ($P_{in}$) polarization were calculated via DFT within the local density approximation (Supplementary Note 4), which yields $P_{out} = 78.1\ \mu C\ cm^{-2} = 0.049\ e\ Å^{-2}$ and $P_{in} = 59.1\ \mu C\ cm^{-2} = 0.037\ e\ Å^{-2}$. The local polarization at point (*x,z*) inside this triangular area can be decomposed to the in-plane $P_{\parallel}(x, z)$ (*x*-component) and out-of-plane $P_{\perp}(x, z)$ (*z*-component). The net in-plane dipole moment can then be estimated by integrating $P_{\parallel}(x, z)$ over the volume of the flux-closure domain $V_{PZT}$. We thus deduced the in-plane dipole per unit length as:

$$p_{\parallel} = \frac{\int P_{\parallel}(x,z)dV_{PZT}}{l} = \iint P_{\parallel}(x,z)\mathrm{d}x\mathrm{d}z = \frac{1}{4}w \times h \times P_{in} = 6.93\ e. \tag{1}$$

Here we assumed the maximum width *w* and depth *h* of the triangle domain to be 2.5 and 3 nm, respectively, based on the TEM result[35], and *l* is the lateral extension of the DW.

We then considered the polar property of 1L H-MoS$_2$, which belongs to the $D_{3h}$ point group. The polar displacement in the unit cell can generate three equal polarizations along those three polar directions, leading to zero net polarization. However, when one of the polar axis is coupled to a neighboring dipole, the rotational symmetry is lifted. Using DFT, we estimated the polarization of MoS$_2$ along one polar direction to be $P_{MoS_2} = 85.5\ \mu C\ cm^{-2}$ (Supplementary Note 4). Using the thickness of 1L MoS$_2$ of $h_1 = 3.11\ Å$, we obtained the dipole moment per unit length for the area above the flux-closure DW in PZT (Fig. 3b):

$$p_{MoS_2} = P_{MoS_2} \times h_1 \times w = 6.65 \times 10^{-19}C \approx 4.15\ e, \tag{2}$$

which is on the same order of $p_{\parallel}$ estimated for the DW in PZT (Eq. 1). While the precise value of the polarization may vary, this simple model naturally explains the major features of our observation. When one of the polar axis of MoS$_2$ is aligned with the in-plane polarization of PZT at the DW regions, as for the odd DWs, their excited interfacial SH dipole fields are coherently coupled[37,38], leading to significantly enhanced SHG response that is linearly polarized along the polar axis. For the anti-aligned even DWs, where these two SH dipole fields cancel each other, the SHG intensity is strongly suppressed. For the vertical ([010]) DWs, on the other hand, $p_{\parallel}$ is not coupled to any of the polar axes of MoS$_2$. The SHG responses of PZT and MoS$_2$ remain to be independent, and we only observe the weak SHG from PZT that is filtered by the MoS$_2$ top layer.

To further test the proposed scenario based on the interfacial polar coupling between MoS$_2$ and the DW, we created square domains on PZT with different stacking angles ($\theta$) with respect to the same MoS$_2$ top layer. Figure 3c shows the PFM phase image of four square-shaped domain structures written at different scanning directions, which are rotated by $\theta = 0°, 15°, 30°$, and $45°$ relative to *x*-axis in the clockwise direction. For bare PZT, the SHG response is uniform at all DW regions, independent of their orientations (Fig. 3d). We then transferred a lL MoS$_2$ flake on top of this area, with the *a*-axis (zigzag orientation) aligned along *x*-direction. As shown in Fig. 3e, without an analyzer, the stacking angle between MoS$_2$ and DW has a clear impact on the reflected

SHG intensity, suggesting that the heterointerface acts as an unconventional light polarizer.

The net SHG response for each of these DWs can be well modeled using the nonlinear electromagnetic theory, considering the second-order nonlinear optical susceptibility tensors for the MoS$_2$/PZT heterointerface. As the thickness of MoS$_2$ and the depth of the flux-closure region at PZT DW (*h*) are well below the optical wavelength, the susceptibility tensor (or the contracted *d*-tensor) of the composite system equals to the sum of the adjacent layers: $d_{interface}^{(2)} = d_{MoS_2}^{(2)} + d_{DW}^{(2)}$, where $d_{MoS_2}^{(2)}$ and $d_{DW}^{(2)}$ are the *d*-tensors for MoS$_2$ and DW, respectively. For 1L MoS$_2$ with $D_{3h}$ point group symmetry, the second-order *d*-tensor can be expressed as[29]:

$$d_{MoS_2}^{(2)} = \begin{pmatrix} 0 & 0 & 0 & 0 & 0 & d_{16}' \\ d_{21}' & d_{22}' & 0 & 0 & 0 & 0 \\ 0 & 0 & 0 & 0 & 0 & 0 \end{pmatrix}, \tag{3}$$

where $d_{21}' = d_{16}' = -d_{22}' = d_{MoS_2}$. For the tetragonal PZT thin films with 4 mm point group symmetry, the *d*-tensor can be written as[39]:

$$d_{PZT}^{(2)} = \begin{pmatrix} 0 & 0 & 0 & 0 & d_{15} & 0 \\ 0 & 0 & 0 & d_{15} & 0 & 0 \\ d_{31} & d_{31} & d_{33} & 0 & 0 & 0 \end{pmatrix}, \tag{4}$$

where $d_{15} = d_{31}$, and $d_{33} \approx 0.9d_{15}$. The tensor elements for PZT were obtained by averaging the calculated and experimental values[31,39]. In our work, the crystallographic axes of PZT coincide with the experimental reference frame (*x*, *y*, *z*), where the [001] orientation of PZT is along the *z*-axis (Fig. 1a). We first considered a square $P_{down}$ domain embed in a $P_{up}$ region in PZT (Fig. 3c), with the horizontal ([100]) DW aligned with the *a*-axis of MoS$_2$ (stacking angle $\theta = 0°$). The interfacial composite tensors at the $P_{up}$ and $P_{down}$ domains are given by:

$$d_{P_{up}}^{interface} = \begin{pmatrix} 0 & 0 & 0 & 0 & d_{15} & d_{MoS_2} \\ d_{MoS_2} & -d_{MoS_2} & 0 & d_{15} & 0 & 0 \\ d_{15} & d_{15} & d_{33} & 0 & 0 & 0 \end{pmatrix},$$
$$d_{P_{down}}^{interface} = \begin{pmatrix} 0 & 0 & 0 & 0 & -d_{15} & d_{MoS_2} \\ d_{MoS_2} & -d_{MoS_2} & 0 & -d_{15} & 0 & 0 \\ -d_{15} & -d_{15} & -d_{33} & 0 & 0 & 0 \end{pmatrix}. \tag{5}$$

As the lateral polarization for the flux-closure domain is comparable with the bulk polarization of PZT, we obtained the *d*-tensors for the four DWs (Top–DW, Bottom–DW, Left–DW, and Right–DW) via a rotation matrix transformation (Supplementary Note 5)[31]:

$$d_{Top-DW}^{(2)} = -d_{Bottom-DW}^{(2)} = \begin{pmatrix} 0 & 0 & 0 & 0 & 0 & d_{15} \\ d_{15} & d_{33} & d_{15} & 0 & 0 & 0 \\ 0 & 0 & 0 & d_{15} & 0 & 0 \end{pmatrix},$$
$$d_{Left-DW}^{(2)} = -d_{Right-DW}^{(2)} = \begin{pmatrix} d_{33} & d_{15} & d_{15} & 0 & 0 & 0 \\ 0 & 0 & 0 & 0 & 0 & d_{15} \\ 0 & 0 & 0 & 0 & d_{15} & 0 \end{pmatrix}. \tag{6}$$

The interfacial composite tensors at the DWs can thus be expressed as:

$$d_{\text{Top}-\text{DW}}^{\text{interface}} = \begin{pmatrix} 0 & 0 & 0 & 0 & 0 & d_{\text{MoS}_2} + d_{15} \\ d_{\text{MoS}_2} + d_{15} & -d_{\text{MoS}_2} + d_{33} & d_{15} & 0 & 0 & 0 \\ 0 & 0 & 0 & d_{15} & 0 & 0 \end{pmatrix},$$

$$d_{\text{Bottom}-\text{DW}}^{\text{interface}} = \begin{pmatrix} 0 & 0 & 0 & 0 & 0 & d_{\text{MoS}_2} - d_{15} \\ d_{\text{MoS}_2} - d_{15} & -d_{\text{MoS}_2} - d_{33} & -d_{15} & 0 & 0 & 0 \\ 0 & 0 & 0 & -d_{15} & 0 & 0 \end{pmatrix},$$

$$d_{\text{Left}-\text{DW}}^{\text{interface}} = \begin{pmatrix} d_{33} & d_{15} & d_{15} & 0 & 0 & d_{\text{MoS}_2} \\ d_{\text{MoS}_2} & -d_{\text{MoS}_2} & 0 & 0 & 0 & d_{15} \\ 0 & 0 & 0 & 0 & d_{15} & 0 \end{pmatrix},$$

$$d_{\text{Right}-\text{DW}}^{\text{interface}} = \begin{pmatrix} -d_{33} & -d_{15} & -d_{15} & 0 & 0 & d_{\text{MoS}_2} \\ d_{\text{MoS}_2} & -d_{\text{MoS}_2} & 0 & 0 & 0 & -d_{15} \\ 0 & 0 & 0 & 0 & -d_{15} & 0 \end{pmatrix}.$$

(7)

Furthermore, we derived the explicit expressions of interfacial SHG tensors at the four DWs as a function of stacking angle $\theta$, which are given in Supplementary Eq. 5. The SH dipole field $\boldsymbol{P}_{\text{interface}}^{2\omega}$ is given by the product of $d$-tensors and the fundamental field, and the SHG intensity at the 1L MoS$_2$/PZT interface is given by:

$$I_{\text{SHG}}(\varphi = 0^{\circ}) \approx \left| \boldsymbol{P}_{\text{interface}}^{2\omega}(\varphi = 0^{\circ}) \right|^2. \qquad (8)$$

To simplify the calculation, we assumed that the maximum SHG intensities from MoS$_2$ and flux-closure domain of PZT are the same, which is reasonable given their closely matched dipole moments. Figure 3f shows the simulated SHG results, which capture well the features of the SHG tailoring effect for all stacking angles (Figs. 2e and 3e). Supplementary Table 2 lists a detailed comparison between the experimental and modeling results, which shows an excellent agreement, yielding strong support to the scenario for the interfacial polar coupling between MoS$_2$ and PZT DW.

Comparing the results shown in Fig. 2b, e, it is clear that the lateral polarization of PZT DW can replace an optical analyzer to provide efficient filtering of the light polarization for the SHG signal of MoS$_2$. It further enhances or quenches the SHG intensity for the selected light polarization depending on the underlying DW chirality. Compared with the existing optical filter technologies, which are macroscopic in terms of dimensions, time consuming in terms of optical set-up, and cannot be programmed at the nanoscale, the MoS$_2$/PZT heterostructure has the distinct advantages in terms of size scaling and being nanoscale reconfigurable, offering the opportunities to achieve on-chip generation and smart filtering of SHG signals for nano-optics.

### Transmitted SHG response of 1L MoS$_2$/PZT heterostructures.

While bare PZT domains (Supplementary Fig. 3) and MoS$_2$ on Gel-Films (Supplementary Fig. 4) exhibit similar SHG responses in the reflection and transmission modes, the MoS$_2$/PZT heterostructure reveals qualitatively different SHG tailoring effects in these two detection modes. Figure 4a displays the transmitted SHG image of the same domain structure shown in Fig. 2e. Overall, the maximum intensity for the transmitted light is comparable or lower than that for the reflection mode depending on PZT thickness, as the signal is collected through the oxide layers (Supplementary Fig. 4g, h). In sharp contrast to the reflected SHG image, a clear signal contrast emerges between the $P_{\text{up}}$ and $P_{\text{down}}$ domains, rather than at the DWs, in the

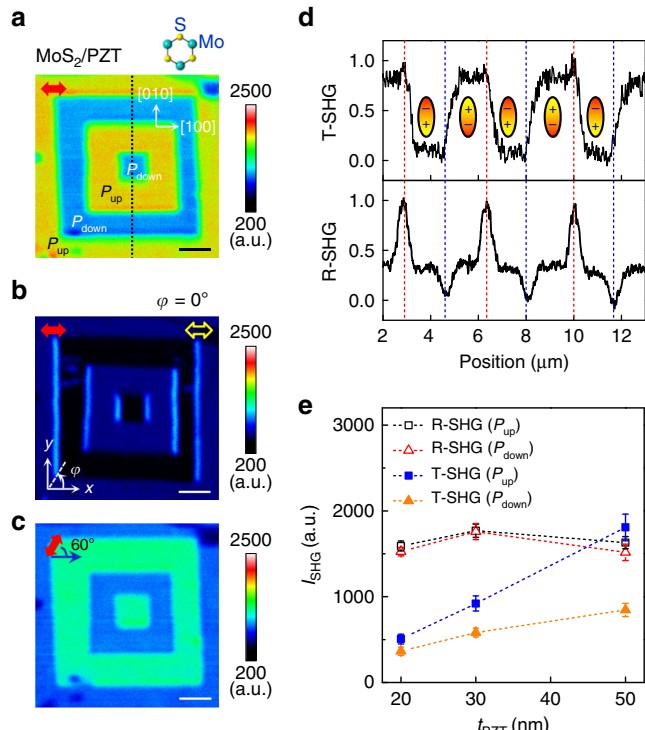

**Fig. 4 Transmitted SHG response of 1L MoS$_2$/PZT interface. a–c** SHG mapping in transmission mode of the same MoS$_2$/PZT sample shown in Fig. 2 taken with no analyzer applied (**a**, **c**) and with a horizontal analyzer applied (**b**). The red solid (yellow open) arrows mark the incident light (analyzer) polarization. The crystalline orientations of PZT and MoS$_2$ are shown as insets in **a**. The scale bars are 2 μm. All images were taken at the excitation laser power of 20 mW. **d** Normalized SHG intensity profiles obtained in the transmission (T-SHG) and reflection (R-SHG) modes along the black dotted line in **a**. The dashed lines serve as the guide to the eye. **e** Averaged SHG intensity as a function of PZT thickness taken on the $P_{\text{up}}$ (squares) and $P_{\text{down}}$ (triangles) domains in both reflection (open symbols) and transmission (solid symbols) modes at excitation laser power of 20 mW.

transmission mode. Only when a horizontal analyzer is applied does the SHG signal for the vertical ([010]) DWs become appreciable (Fig. 4b), similar to that observed in the reflected mode (Fig. 2h). We also note that the relative strength of the SHG intensity between the regions of the $P_{\text{up}}$ and $P_{\text{down}}$ domains depends on the polarizer angle (Fig. 4c). Figure 4d shows the cross-sectional SHG signal profiles for the same region in both transmission (Fig. 4a) and reflection (Fig. 2e) modes. For comparison, the signals are normalized to the intensity difference between the even ($I_{\text{even}}$) and odd ($I_{\text{odd}}$) horizontal DWs, defined as $(I - I_{\text{even}})/(I_{\text{odd}} - I_{\text{even}})$. It clearly illustrates that the signal contrast is either tailored by the uniformly polarized domains or the DWs in these two detection modes.

To understand the origin for this ferroelectric polarization-dependent SHG response, we quantitatively compared the signal intensity in 1L MoS$_2$/PZT heterostructures with different PZT layer thicknesses for both detection modes (Supplementary Fig. 7). As shown in Fig. 4e, the intensity of the reflected SHG signal does not exhibit apparent dependence on PZT thickness, consistent with its interfacial origin. The transmitted light intensity, on the other hand, increases monotonically with the layer thickness of PZT for both $P_{\text{up}}$ and $P_{\text{down}}$ domains, with the signal at the $P_{\text{up}}$ domain approaching the intensity for the reflected signal in the heterostructure with 50 nm PZT, which

suggests that this tailoring effect is related to the bulk state of PZT. Varying the focus plane for the transmission image shows that the SHG signal is fully attenuated in the STO substrate (Supplementary Fig. 8), confirming that the relevant dielectric layer for this tuning effect is indeed PZT. The observed film thickness dependence is opposite to what is expected owing to light absorption in a non-transparent dielectric layer. We thus speculate that the tailoring of the transmitted SHG signal originates from a possible cavity effect of the PZT layer through constructive interference among multiple reflections. To verify this scenario, however, requires working with much thicker PZT films, ideally larger than half wavelength of the SHG light ($\lambda_{air}/2n_{PZT} \approx 83$ nm, with $n_{PZT} \approx 2.4$). This is challenging as the PZT films thicker than 50 nm tend to relax the epitaxial strain through forming $a$-domains[32], making the local polarization orientation not well defined. The SHG contrast between the regions on the $P_{up}$ and $P_{down}$ domains, on the other hand, depends sensitively on the light polarization and can reverse the relative strength (Fig. 4a, c). The difference is thus likely phase related rather than due to the doping difference and may originate from the polarization-dependent surface reconstruction in PZT thin films[36].

## Discussion

In summary, we report an interface-driven nonlinear optical filtering effect in 1L MoS$_2$/ferroelectric heterostructures. The tailoring effect for the reflected SHG signal is solely determined by the polar symmetry of MoS$_2$ and PZT DW. The transmitted SHG signal, in sharp contrast, is sensitively tuned by the out-of-plane ferroelectric polarization rather than the DW, which is attributed to the bulk state of PZT. Our study points to a new material platform for the functional design of novel interfacial optical response via ferroelectric domain patterning. This approach can be widely applied to vdW materials and heterostructures with broken inversion symmetry, paving the way for achieving nanoscale electrically programmable optical filtering applications.

## Methods

**Preparation and characterization of epitaxial PZT**. We deposited 20–50-nm-thick epitaxial PZT films on 10 nm LSMO-buffered (001) SrTiO$_3$ substrates (5 mm × 5 mm × 0.5 mm) via off-axis radio frequency magnetron sputtering. The LSMO layer was deposited at 650 °C in 120 mTorr process gas composed of Ar and O$_2$ (ratio 2:1). We then deposited the PZT layer in situ at 490 °C at 150 mTorr process gas (Ar:O$_2$ = 2:1). The PZT films are $c$-axis oriented with out-of-plane polar axis (Supplementary Fig. 1a). AFM images show smooth surface morphology with 2–3 Å surface root mean square roughness.

**Preparation of MoS$_2$/PZT heterostructure**. 1L and few-layer MoS$_2$ flakes were mechanically exfoliated on elastomeric films (Gel-Film® WF × 4 1.5 mil from Gel-Pak) from bulk single crystals. Selected flakes were transferred on top of the patterned domain structures using an all-dry transfer technique[40]. The Gel-Film with exfoliated MoS$_2$ was flipped upside down and anchored with a high-precision XYZ manipulator. The PZT sample was placed on a rotatable hot plate. We then aligned the MoS$_2$ sample with the patterned domains under an optical microscope with submicron precision. The uncertainty of stacking angle $\theta$ is 2°–6°. The details of the sample alignment during transfer can be found in Supplementary Note 2 and Supplementary Fig. 4.

**PFM measurements**. PFM studies were carried out using a Bruker Multimode 8 AFM. The measurements were performed in contact mode using conductive PtIr-coated tips (SCM-PIT, spring constant $k$ of 1–5 Nm$^{-1}$, resonant frequency $f_o$ of 60–100 kHz). The coercive voltage of the PZT films is about $+2$ V ($-3$ V) for the $P_{up}$ ($P_{down}$) state (Supplementary Fig. 1b). For domain writing, a ±7 V DC bias was applied to the AFM tip while scanning, and the LSMO bottom layer was grounded. For imaging, an AC voltage of 0.5 V was applied at close to the contact resonant frequency. The resolution of PFM is about 5 nm for our experimental set-up[41], which cannot resolve the intrinsic DW width.

**Raman and PL measurements**. Raman and PL measurements were performed on a micro-Raman system (Renishaw InVia plus, Renishaw) at room temperature. An Ar$^+$ laser of about 200 μW was focused to a 1 μm beam spot on the sample at normal incidence. Both Raman and PL spectra were collected in reflection mode through a ×50 objective lens with an accumulation time of 10 s.

**SHG measurements**. The experimental set-up for SHG imaging is shown in Fig. 1a. The laser source for SHG microscopy is provided by a mode-locked Ti: Sapphire fs laser (MaiTai DeepSee HP, SpectraPhysics) with a fixed wavelength of 800 nm, duration of 100 fs, total output power of 2.95 W, and repetition rate of 80 MHz. The laser beam passed a polarizer with normal incidence and then was guided by mirrors into a laser scanning microscope (LSM). In the LSM, the laser beam was linearly focused onto the sample surfaces using a water-immersed Olympus objective lens (1.05 NA, ×25). To avoid water contact, a 0.17-mm thin glass cover slide was placed above the sample surface, forming a thin air gap between the sample and the cover slide. The sample was place on a glass slide (1 mm), lying in the x–y plane, which is placed above the 1" diameter stage opening. The incident light was transversely polarized and directed to the sample surface along $-z$ direction, and the excited SHG signals were collected in both reflection ($+z$) and transmission ($-z$) geometries by photomultiplier detectors (photomultiplier tubes (PMTs)). Before the SHG signals enter the PMTs, the excitation laser beam was filtered out by an IR cut filter (OD >4 @ 692–1100 nm). A long working distance (WD) condenser (NA 0.8/WD 5.7 mm) was used for the transmission signal collection. In the LSM, the transmission and reflection modes share the same focus plane using the same type of objective lens (NA 1.05/water immersed/WD 2.0 mm, ×25), so both measurements can be performed simultaneously. The focusing to the MoS$_2$/PZT interface was performed through the reflected mode. The signal was first collected with no analyzer inserted and then with an analyzer inserted in different angles with respect to the polarizer orientation. The analyzer is a Thorlab LPVISE100-A with operating wavelength range of 400–700 nm. The band-pass filter used for SHG imaging is a Semrock FF01-390/40 ($T_{avg}$ >93% @ 370–410 nm, center wavelength of ~390 nm, and bandwidth of ~40 nm). The diffraction limit of the excitation laser beam (spot size) was estimated to be $\lambda/2NA = 380$ nm. Owing to the second-order nonlinearity of the SHG light, the spatial resolution was estimated to be ~300 nm.

The SHG mapping plots, unless otherwise specified, are the raw data collected by the PMTs without modification. During the SHG measurements, the PMTs in the LSM were set on the photon count mode, so the responses of the PMTs are proportional to the number of actual SHG photons detected by the PMTs but are not calibrated to the light intensity in units of W/m². The SHG mapping results were expressed in terms of arbitrary units and were proportional to the actual SHG intensity detected by the PMTs. The intensity level can be directly compared if they were taken at the same laser power.

## Data availability

All relevant data that support the findings of this study are available from the corresponding authors upon request.

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

## Acknowledgements

We thank Anthony Starace and Ziliang Ye for helpful discussions and Chengyiran Wei and Kun Wang for technical assistance. This work was primarily supported by the U.S. Department of Energy (DOE), Office of Science, Basic Energy Sciences (BES) under Award No. DE-SC0016153. Additional support is provided by the NSF Nebraska Materials Research Science and Engineering Center (MRSEC) Grant No. DMR-1420645 (PZT growth and theoretical modeling). Y.L. acknowledges the support from NSF Award No. CMMI 1826392 and ONR Award No. N00014-15-C-0087. The research was performed in part in the Nebraska Nanoscale Facility: National Nanotechnology Coordinated Infrastructure and the Nebraska Center for Materials and Nanoscience, which are supported by the National Science Foundation under Award ECCS: 1542182, and the Nebraska Research Initiative.

## Author contributions

X. Hong, D.L., and Z.X. conceived the project and designed the experiments. X. Hong supervised the project. L.Z. and Y.H. prepared the PZT thin films. J.S. and D.L. carried out the surface and structural characterizations of PZT. Z.X., D.L., and H.C. prepared the MoS₂/PZT heterostructures. X. Huang, D.L., and Y.L. contributed to the Raman, PL, and SHG studies. D.L. and Z.X. carried out the PFM studies. D.-F.S. and E.Y.T. performed theoretical modeling of the polar states. D.L. and X. Hong wrote the manuscript. All authors discussed the results and contributed to the manuscript preparation.

## Competing interests

The authors declare no competing interests.
