## [Peer Review File · Nature Communications]

Editorial Note: Parts of this Peer Review File have been redacted as indicated to remove references to third-party material for which we do not have permission to publish.

Reviewers' Comments:

Reviewer #1:

Remarks to the Author:

Nature Commun. Referee

Polar Coupling Enabled Nonlinear Optical Filtering at MoS₂/Ferroelectric Heterointerfaces

This paper describes second harmonic (SH) responses at the heterointerface between a non-centrosymmetric monolayer MoS₂ and a ferroelectric PZT thin film with tailored domain structure. The authors insist that the interfacial polar coupling between two substances enhances or quenches SHG intensities and that this heterostructure could exhibit a filtering effect, which does occur at the domain walls and not at domains.

The referee thinks that the paper needs deeper considerations and further experiments to clarify the origin of these experimental results. These are summarized as follows:

1. The authors claim that the SHG responses originate from the polar-coupling between MoS₂ and PZT. However, the referee suspects that these can be simply explained by the mere interference effect, i.e., coherent addition or subtraction of SH waves from PZT and MoS₂: When SH waves from the two materials are in-phase, they are added, while they are subtracted in the case of out-of phase. See references (1-2). For proving or denying this possibility, it is necessary to make additional experiments changing thicknesses of MoS₂ and/or PZT films, or replacing MoS₂ with another polar material. It would be also effective to calculate the phases of SH waves using the anisotropy and dispersion of the refractive indices of both materials.

(1) S. Kurimura et al, "Application of the second harmonic generation microscope to nondestructive observation of periodically poled ferroelectric domains in quasi-phase matched wavelength converters," J. Appl. Phys. 81, 369 (1997).

(2) J. Kaneshiro et al, Visibility of inverted domain structures using the second harmonic generation microscope: Comparison of interference and non-interference cases, J. Opt. Soc. Am. B27, 888 (2010).

2. Specify the spatial resolution of the SHG microscope used in this experiments. The thickness of domain walls is estimated to be about 1.5 μm from Fig. 1c but is it the true width or the artifact due to the convolution effect of a true domain wall and the microscope resolution? The authors should make a comment on it.

3. The authors should explain more precisely how to put an MoS₂ film exfoliated from a bulk crystal on PZT. Is it attached on PZT as atomic-scale contact? How is it estimated? Is there no space charges effect or depolarization effect at the interface?

4. The "zigzag axis" (p4-2) and the "armchair direction" (p6-17) are terms in popular parlance. It is preferable to be described with the crystallographic axes.

5. What do the authors want to conclude from the result of PL spectra (Fig. 1e and f). Give briefly concrete and specific facts related to this hetero-system.

6. Put the orientation of domain walls of PZT on Fig. 1c. The thicknesses of MoS₂ and PZT should be also given in the early part of the manuscript.

In conclusion, the referee is negative to the publication of this manuscript in Nature Communications.

Reviewer #2:

Remarks to the Author:

In this work, the authors reported an unconventional nonlinear optical filtering effect resulting from the coupling between monolayer MoS₂ and ferroelectric oxide thin film. The second harmonic generation response at the hetero-interface is either substantially enhanced or almost entirely quenched by an underlying ferroelectric domain wall depending on its chirality. Although the effect is interesting, it is not well interpreted and the so called "symmetry coupling" is misleading, making a simple phenomenon more confusing to the readers. Specific issues are listed below:

1. The assumed polarization configuration in Fig. 3a may not be correct. Various experiments had proven that the 180° domain walls are very sharp (within one unit cell in most cases), the Science article by Jia et al about the continuous polarization rotation in a 180° domain wall might be misleading, since it involves elastic deformation in the domain wall region, making it a higher energy configuration. Hence, such continuous polarization rotation should not occur often in the system. The actual polarization configuration created by the PFM tip should follow the electric field lines generated by the electrode configuration, which could be simulated using finite element software, such as ANSYS, COMSOL, etc.

2. Since the purpose of this work is to prove polarization coupling between the PZT film and monolayer MoS₂, the experiments could be performed more conveniently if one could just use an interdigital electrode on one surface of the film, then pole the thin film into a periodic alternating in-plane polarization pattern. One could control the in-plane polarization strength by applying different levels of electric field. It would be better quantified and more convincing than the first principles estimate for the in-plane polarization of a domain wall, since the first principles calculations could not give accurate quantitative results. Why introduce domain walls into the picture if one just wants to make the polarization of monolayer MoS₂ to couple with an in-plane polarization field?

3. When the monolayer MoS₂ is deposited on a substrate with an existing polarization pattern, the polarization of the two would either enhance or cancel each other, which is nothing special about. Although the experimental procedure is well executed in this work, there is no new concept generated and I am also having a hard time to see the purpose of this research effort. Obviously, it would be a poor optical filter since the PZT film is not a transparent material. There are many other ways to make better optical filters, the authors need to show the advantages of such filters over existing filters in order to let the readers see the value of this research.

4. "The SHG signal is solely mediated by the interfacial polar symmetry" is not an accurate statement. In my opinion, the SHG is still mediated by the combined polarization of the film and the monolayer MoS₂, not by the symmetry. Symmetry does not "couple".

With the above arguments, I do not recommend publication of this work in Nature Communications.

Reviewer #3:

Remarks to the Author:

1. KEY RESULTS

The paper investigates second harmonic generation (SHG) at the interface between a 50nm-thin ferroelectric oxide film (PZT) structured with up and down (180°) domains, and a monolayer (1L) of transition metal dichalcogenide (MoS₂).

The SHG response in reflection highlights the interplay of the in-plane components of the spontaneous polarization at the interface between the MoS₂ layer [Phys. Rev. B 87, 161403 (2013)] and Neel-like domain walls (DWs) in PZT [Nature Comms 8, 15768 (2017)]. As a result, the SHG signal arising from the latter is either enhanced or suppressed, depending on the chirality of the underlying DW and its orientation with respect to the MoS₂ polarization. The experimental observations are complemented by a density functional theory analysis and justified by symmetry arguments. SHG studies are also provided for the same system in transmission. There is a qualitatively different response observed, where out-of-plane polarization components of PZT appear to play a non-negligible role.

2. NOVELTY & INTEREST

The main novelty of the paper lies in probing the symmetries of the TMDC /PZT-DW interface using SHG techniques. The symmetries of each component per-se are well-known and have been studied by SHG. The MoS₂ monolayer exhibits a polar in-plane axis, while the ferroelectric oxide (here grown along z) exhibits 180° domains, with additional chiral structure at DW, as recently revealed by SHG [Nature Comms 8, 15768 (2017)]. Nevertheless, the combination of these two systems

had not been studied before from the SHG point of view. The main SHG results here show that the in-plane polarization components of MoS2 and PZT-DW give rise to a combined nonlinear optical response at the interface, which is in principle engineerable locally through its constituents. The results are interesting also in that the combined polarization of MoS2-1L and PZT-DWs (or equivalently of their nonlinear optical tensors) provides a means to identify the chirality of the latter, not easily accessible by measurements on the ferroelectric oxide system on its own. Hybrid systems of 2D TMDCs and domain-patterned ferroelectric oxides are interesting for their electronic and optoelectronic properties [Nature 560, 336 (2018), ACS Nano 9, 8089 (2015), PRL 236801 (2017), ACS Omega 1, 1075 (2016)] and this paper demonstrates potentially interesting avenues for the nonlinear optics community.

3. VALIDITY

- a) The work is convincing in terms of the SHG results in reflection and of their interpretation, although it would be suitable to complement them with a more comprehensive SHG analysis.
- b) The argumentations provided for the SHG results obtained in transmission (Fig. 4), including statements on the coupling of the out-of-plane polarization of PZT to the polar axis of MoS2 (e.g. line 17 on page 9) appear less convincing and need thorough revision. The discussion provided on pages 9-10 has also a number of flaws, such as: 1) the refractive index used for PZT ($n=10$) is clearly wrong: it should be ~ 2.4 (the authors here incorrectly use the value of the dielectric constant at low frequencies to extrapolate the refractive index at optical frequencies); 2) the focusing action of the PZT layer cannot be used to justify a better collection efficiency, since Snell's law applies also for the bottom interface (PZT/air), unless a different collection scheme is used?; 3) for transverse EM waves propagating along the z-axis, all SH contributions from the z-polarization (P_{out}/P_{in}) should be zero in the far field
– see also more detailed comments in the attached file.

4. DATA, METHODOLOGY & STATISTICAL ANALYSES

- a) It would be appropriate to include a more thorough analysis of the results of Fig. 2-3 in terms of SHG theory and composition of the d-matrices, which is at the moment lacking in the main text and only briefly touched upon in Suppl. Information.
- b) The authors should provide more detailed information on the SHG experimental setup and measurement conditions (e.g. spot size and focusing), in addition to what is currently provided on p. 12. This will also be useful to understand the SHG transmission results.
- c) As done for PZT (Fig. S3), it would be good to provide measurements for SHG on MoS2-alone and MoS2 on +/-z PZT (as references), both for transmission and reflection. Current information provided by Fig.S5 is limited (and not very clear) in this respect.
- d) A further comment on the procedure used to align the zig-zag axis of the MoS2 flake to the PZT DW and an estimate of the associated uncertainty on the stacking angle (θ) for all shown data, including theory-experiment comparisons in table S2, should be included.
- e) Arbitrary units are used throughout for the SHG maps. Please provide a common reference, or absolute values, allowing to compare the intensity levels across different measurements, in particular those of Fig. 2a-h, Fig. 4a-c, Fig. S3a-h.
– see also more detailed comments in the attached file.

LIST OF TECHNICAL COMMENTS

(with reference to manuscript page numbering)

Page 3 -lines 19-20. When introducing the PZT material system, its crystallographic axes and the experimental reference frame should also be defined (this is done at some point in the Suppl. Information, but it is good to have this information at hand for all data. It would also eliminate possible ambiguities in using the terms 'parallel' and 'horizontal', 'vertical' in the text, see below)

Page 4 – lines 12-14 and Suppl Information Fig. S5c. As discussed in the comments for authors: further measurement data and related information should be provided concerning SHG on MoS₂-alone, MoS₂ on +z PZT and MoS₂ on -z PZT, both in transmission and reflection SHG, as references. If measurements on stand-alone MoS₂ films are not available, the role of the substrate and its polarity in both transmission and reflection measurements should be investigated and commented upon. Please note also that the text refers to MoS₂ flakes on gel films, while the title of Fig. S5c mentions MoS₂ on PZT.

Page 4, lines 14-15. *“As the incident light is along the polar axis...”*, please reformulate better, e.g.: *“As the incident light is A TRANSVERSELY-POLARIZED (x,y) ELECTROMAGNETIC WAVE PROPAGATING along the polar axis...”* (if this guess is correct)

Page 5, lines 2-3. *“ the SHG signals in the transmitted mode exhibit qualitatively similar behaviors as in the reflected mode (Supplementary Fig. S3).”* Please provide also scales for the signal intensities to enable a comparison (instead of a.u. maybe normalize to the same value).

Page 5, lines 14-15. Please modify the following misleading statement: *“clearly showing that the interfacial SHG signal is not a simple summation of the signal strength from each constituent layer. The horizontal DW, more surprisingly, exhibit alternating ...”*. In electromagnetic terms the interfacial SHG signal is equal to the sum of the fields arising from each constituent layer, as indeed seen in the experiments. Furthermore, from a SHG perspective, the results obtained on the interfacial layer are not too surprising in terms of the known symmetries of MoS₂ and Neel-like DWs in PZT (ref. 25) and of the way in which they should compose in terms of SH polarization. Without resorting to DFT calculations, the results can be interpreted in terms of SH polarization components and related SHG tensors. This is briefly touched upon in Supplementary information but is worth a more thorough explanation and analysis as it provides the underlying theory to interpret all experimental results in Fig. 2 and 3. Specifically, for the ultra-thin layers (\ll optical wavelength) considered in the paper, basic nonlinear electromagnetic theory would predict the d-matrix (SHG tensor) of the composite system ($d_{ij}^{\text{interface}}$ of p. 8 in Suppl. Inf.) to be equal to the sum of the d-matrices of the adjacent layers, i.e. d-MoS₂ (Suppl Information, p. 7) and d-DW (Suppl Information, p. 8), just because the FF field is the same and the SH polarization vectors from the MoS₂ and PZT-DW contributions do add up. This implies that the nonlinear matrix components d_{yx} shall either add up constructively (in the case of: $d_{\text{MoS}_2} + d_{\text{Top-DW}}$) or destructively (in the case of: $d_{\text{MoS}_2} + d_{\text{Bottom-DW}}$) for the horizontal DWs, explaining the results in fig. 2f. With the same methodology, looking at the d_{xx} components, one can easily justify why the MoS₂ layer does affect SHG from the vertical DWs (Fig. 2h). Related derivations and explanations should be included in the Supplementary Information and used in the text as they provide the general framework for a correct interpretation of the SHG data.

Page 5, lines 15-22 and Suppl. Information. The alternating enhancement/suppression of the SHG signals can be easily explained with the d-tensors. Please include the reference frame in the main text, expand the analysis of Supplementary Information providing the explicit expression of the

overall tensors for the (MoS₂+DW) and (MoS₂+PZT) combinations and refer to the former to clarify the interplay of the tensor symmetries and the observed SHG results (Fig. 2) in the main text. In doing so, it would also be appropriate to differentiate among the different components of the d-matrices (instead of using just one coefficient, *A* or *B* for each material), especially since they can be substantially different with non-negligible impact on the SHG amplitudes for different polarization configurations (e.g. in PZT the difference between d₃₁, d₁₅ and d₃₃ components yields differences of more than one order of magnitude in SHG intensities).

Page 6, line 18. “parallel” in this context means probably ‘horizontal’ (DW). In any case it is better to remove this ambiguity by introducing and using a common x,y,z reference frame (1st comment).

Pages 7-8. Theoretical modelling. As mentioned in the comments to Authors, apart from DFT simulations on the polar coupling, concerning ultimately only one direction (//) the symmetries of the problem should be introduced and analysed in the more general theoretical framework of SHG tensors (d-matrices). This applies to the expressions of the d-matrices to be used in the analysis of the data of Fig. 2 and Fig. 3 (and table S2). Hence also the explicit expression of $d_{ij}^{\text{interface}}$ as a function of θ should also be included in Suppl. Information.

Page 9-10. Transmitted SHG response. This part requires thorough revision.

- The Methods part or Suppl. Information should include:
 - details on the experimental setup used the SHG transmission measurements: what was the optics for signal collection (medium underneath the PZT sample, type of objective, magnification, NA, etc.)
 - information of the bottom facet of the PZT sample (was PZT deposited on another substrate? If so which one & how thick? Which other interfaces are traversed by the signals in transmission mode?)
- The focusing conditions (in particular the spot size) of the fundamental beam should be specified in the methods section (both for transmission and reflection measurements)
- The claims and justifications for the coupling between “the out-of-plane bulk polarization of PZT” and “the polar axis of MoS₂” need to be substantiated with other arguments and further clarifications.
- The notation used in Fig. 4 is misleading, whereby the propagation directions of the fields (normally labelled by the wavevector \mathbf{k}) are labelled as electric (\mathbf{E} -) fields, which – in the case of ordinary TEM waves – are polarized orthogonally to the former (\mathbf{k}).
- Incorrect statements made on page 10 (highlighted in the comments for authors) should also be removed.

We thank the reviewers for reviewing our manuscript, and appreciate their critical comments and valuable suggestions. The Reviewers have made very positive comments on the novelty and potential scientific and technological interest of this work. Reviewer 2 considers the observed nonlinear optical filtering effect as “*unconventional*” and “*interesting*”. Reviewer 3 comments that: “*The main novelty of the paper lies probing the symmetries of the TMDC /PZT-DW interface using SHG techniques... the combination of these two systems had not been studied before from the SHG point of view.*” Reviewer 3 also recognizes that the tailoring/filtering of the interfacial nonlinear optical response “*is in principle engineerable locally through its constituents*”, pointing out that our study “*provides a means to identify the chirality of the latter, not easily accessible by measurements on the ferroelectric oxide system on its own*” and “*this paper demonstrates potentially interesting avenues for the nonlinear optics community.*”

The reviewers, however, have raised some questions/concerns that require: 1) additional experiments for further clarifying the origin of the observed synergy between monolayer (1L) MoS₂ and the domain wall (DW)/polar domain in PbZr_{0.8}Ti_{0.2}O₃ (PZT), 2) detailed discussions for the experimental setup and procedures, and 3) clear elaboration of the potential impact of the work. To address these comments, we carried out experiments on new MoS₂/PZT samples with different layer thicknesses and performed additional SHG modeling based the nonlinear optical susceptibility tensor calculation. We also made extensive revisions to the texts to outline the experimental details and highlight the potential technological impact of the work. These changes have significantly improved the clarity and technical strength of the manuscript, for which we sincerely thank the reviewers' help. Below is the point-by-point response to the reviewers' comments. A complete list of changes can be found at the end of the response letter. These changes are also highlighted in the revised manuscript and Supplementary Information.

Response to Reviewer #1's Comments

Reviewer 1 has raised questions about our proposed mechanism underlying the synergetic effect between monolayer MoS₂ and the DW in PbZr_{0.8}Ti_{0.2}O₃ (PZT), commenting that “*the paper needs deeper considerations and further experiments to clarify the origin of these experimental results*”, and requested more detailed discussion of the experimental conditions and procedures. To address these comments, we have carried out additional experiments and made substantial revisions to the manuscript, as discussed in detail below.

Comment 1.1:

1. The authors claim that the SHG responses originate from the polar-coupling between MoS₂ and PZT. However, the referee suspects that these can be simply explained by the mere interference effect, i.e., coherent addition or subtraction of SH waves from PZT and MoS₂: When SH waves from the two materials are in-phase, they are added, while they are subtracted in the case of out-of phase. See references (1-2). For proving or denying this possibility, it is necessary to make additional experiments changing thicknesses of MoS₂ and/or PZT films, or replacing MoS₂ with another polar material. It would be also effective to calculate the phases of SH waves using the anisotropy and dispersion of the refractive indices of both materials.

(1) S. Kurimura et al, “Application of the second harmonic generation microscope to nondestructive observation of periodically poled ferroelectric domains in quasi-phase matched wavelength converters,” *J. Appl. Phys.* 81, 369 (1997).

(2) J. Kaneshiro et al, *Visibility of inverted domain structures using the second harmonic generation microscope: Comparison of interference and non-interference cases*, *J. Opt. Soc. Am.* B27, 888 (2010).

Reply:

We thank the reviewer for the valuable suggestions. To clarify whether the observed SHG signal is originating from the MoS₂/PZT interface, or due to the interference effect between the individual bulk SHG signals from MoS₂ and the PZT film, we carried out additional experiments on new MoS₂/PZT samples with different MoS₂ and PZT layer thicknesses. The underlying hypotheses are: 1) if the observed SHG filtering effect has an interface rather than bulk origin, the reflected SHG pattern should be independent of the film thickness of PZT, and the strength of the reflected signal should be attenuated in thicker MoS₂ layers; 2) if the SHG signal is driven by the interfacial polar symmetry, the tailoring effect should be also observed in odd-layer MoS₂ thin flakes (noncentrosymmetric), but be absent in even layer MoS₂ (centrosymmetric).

We first investigated the effect of MoS₂ layer thickness on the SHG tailoring effect. Figure R1a shows a mechanically exfoliated MoS₂ flake, where Raman spectra mapping reveals 1 layer (1L), 2L, 3L, 4L, 5L, and thick multi-layer (ML) regions (Fig. R1b). We then patterned a series of rectangular domains on a 50 nm thick PZT thin film (Fig. R1c), and transferred the MoS₂ flake on top of this domain structure. Upon transfer, the *a*-axis of MoS₂ (zigzag orientation) is aligned with the long side of the domains ([100] orientation in PZT). Figure R1d shows the reflected SHG mapping of the MoS₂/PZT sample with no analyzer applied. We observed the following features in this data:

1. The 1L area (right region of the flake) exhibits alternately enhanced or suppressed SHG signal along the horizontal ([100]) DWs, consistent with that in Figs. 2-3 in the main text.
2. The 3L (upper-right) and 5L (center) areas of the flake exhibit similar alternating enhancement and suppression of the SHG signal along the horizontal ([100]) DWs, consistent with the tailoring effect in the 1L flake. The overall signal strength is lower in thicker flakes.
3. For the 2L (upper-left) and 4L (top) areas of the flake, there is no SHG signal emerging from MoS₂. Weak SHG signals were observed at the PZT DWs, which is consistent with that of bare PZT.
4. In the ML region (lower part), the SHG signal from PZT is attenuated with layer thickness, and eventually diminishes.

Fig. R1 | **a**, Optical image of an exfoliated MoS₂ flake with different thicknesses on SiO₂/Si substrate, with **b**, the corresponding Raman spectra taken in different regions. **c**, PFM phase image of a series of

rectangular domains written on a 50 nm PZT thin film. **d**, Reflected SHG mapping of the same MoS₂ sample transferred on top of the PZT domain structure shown in **c**. The red arrow marks the incident light polarization (along x -axis). The scale bars are 10 μm . The dotted lines serve as the guide to the eye. This figure is included as Supplementary Fig. S6.

The absence of SHG at the DW in the even layer MoS₂ clearly shows that the observed SHG tailoring effect indeed originates from the noncentrosymmetric symmetry of MoS₂. The fact that the tuning pattern diminishes gradually with MoS₂ thickness also confirms that it is an interface phenomenon rather than due the interference of bulk signals. We now added a detailed discussion of this study in Section 3 of the Supplementary Information, and included Fig. R1 in the Supplementary as Fig. S6. We also added the following sentences in the 1st paragraph on Page 6:

“The tailoring of reflected SHG signal at the DW is a robust effect and has been observed in multiple 1L MoS₂ samples. Similar tuning pattern is also observed on three-layer and five-layer MoS₂ flakes on PZT, and is absent in bilayer and four-layer MoS₂ (Supplementary Fig. S6). This observation clearly demonstrates that the observed effect originates from the noncentrosymmetric symmetry of MoS₂. The intensity of the reflected SHG signal gradually diminishes in thicker MoS₂, confirming that it is an interface rather than bulk phenomenon.”

To investigate the effect of PZT film thickness, we fabricated 1L MoS₂ flakes on top of 20 nm and 30 nm thick PZT films, and compared the SHG results with those obtained on 50 nm PZT, as shown in Fig. R2. Despite the different PZT thicknesses, all composite structures exhibit qualitatively similar SHG responses, with alternately enhanced or suppressed SHG signal at the horizontal ([100]) DWs and unappreciable SHG contrast at the vertical ([010]) DWs. This result further confirms that the SHG tuning pattern is independent of PZT’s thickness, yielding additional support to its interfacial nature.

Fig. R2 | **a**, Schematic experimental set up. **b-d**, Reflected SHG mapping of 1L MoS₂ on **a**, 20 nm, **b**, 30 nm, and **c**, 50 nm PZT films patterned with square P_{up} and P_{down} domains, with the crystalline orientations of MoS₂ and PZT shown as insets. The scale bars are 3 μm . The dashed lines highlight the DW positions. Part of this figure is incorporated in Supplementary Fig. S7.

We emphasize that the origin of the DW SHG signal is interfacial by nature. For normal incident light, the SHG response implies that the polar structure inside the DW breaks in-plane inversion symmetry. This condition, however, is not satisfied in bulk PZT. The DW in bulk PZT is on the order of a couple of unit cells and does not possess an in-plane component (Ref. 33: Meyer & Vanderbilt, *PRB* **65**, 104111 (2002), Ref. 34: Hlinka & Marton, *PRB* **74**, 104104 (2006)). The chiral rotation of the local dipole, which is critical for forming the in-plane polarization, can only be stabilized at the surface (Fig. 3a) due to the presence of strong depolarization field (Ref. 35: Jia *et al.*, *Science* **331**, 1420 (2011), Ref. 36: Gao *et al.*,

Nat. Commun. **7**, 11318 (2016)). To clarify these points, we included this data and related discussion in the Supplementary Information (Fig. S7 and Section 3), and added the following sentences on pages 7-8:

“We further compared the SHG response of 1L MoS₂ interfaced with 20 nm, 30 nm, and 50 nm PZT films (Supplementary Fig. S7). Despite the different PZT thicknesses, all heterostructures exhibit qualitatively similar SHG responses, with alternating enhancement and suppression of the SHG signal observed at the horizontal ([100]) DWs and unappreciable SHG contrast observed at the vertical ([010]) DWs. This result further confirms the interfacial nature of the DW’s tailoring effect. In fact, the 180° DW in bulk PZT is on the order of a couple of unit cells and does not acquire an in-plane component.^{33,34} The chiral rotation of the local dipole, which is critical for forming the in-plane polarization, can only be stabilized at the surfaces/interfaces (Fig. 3a) due to the presence of strong depolarization field.^{35,36}”

That being said, we have implicitly incorporated the phase of the SHG signal in modeling the coherent coupling of the interfacial response from MoS₂ and in-plane polarization of PZT, which is similar to that discussed in Kurimura *et al* (*JAP* **81**, 369 (1997)) and Kaneshiro *et al* (*J. Opt. Soc. Am. B* **27**, 888 (2010)). To clarify this point, we added the following sentence and these two references (Refs. 37,38) to the last paragraph on Page 9:

“When one of the polar axes of MoS₂ is aligned with the in-plane polarization of PZT at the DW regions, as for the odd DWs, their excited interfacial SH dipole field are coherently coupled,^{37,38} leading to significantly enhanced SHG response that is linearly polarized along the polar axis.”

We also included the details of SHG modeling using the second-order nonlinear optical susceptibility tensors for the composite system in the Supplementary Information (Section 5) and on Pages 10-12 as follows:

“The net SHG response for each of these DWs can be well modeled using the nonlinear electromagnetic theory, considering the second-order nonlinear optical susceptibility tensors for the MoS₂/PZT heterointerface. As the thickness of MoS₂ and the depth of the flux-closure region at PZT DW (h) are well below the optical wavelength, the susceptibility tensor (or the contracted d -tensor) of the composite system equals to the sum of the adjacent layers: $d_{interface}^{(2)} = d_{MoS_2}^{(2)} + d_{DW}^{(2)}$, where $d_{MoS_2}^{(2)}$ and $d_{DW}^{(2)}$ are the d -tensors for MoS₂ and DW, respectively.

... Furthermore, we derived the explicit expressions of interfacial SHG tensors at the four DWs as a function of stacking angle θ , which are given in the Supplementary Information (Eq. S5).

... Figure 3f shows the simulated SHG results, which capture well the features of SHG tailoring effect for all stacking angles (Figs. 2e and 3e). Supplementary Table S2 lists a detailed comparison between the experimental and modeling results, which shows an excellent agreement, yielding strong support to the scenario for the interfacial polar coupling between MoS₂ and PZT DW.”

Comment 1.2:

2. Specify the spatial resolution of the SHG microscope used in this experiments. The thickness of domain walls is estimated to be about 1.5 μm from Fig. 1c but is it the true width or the artifact due to the convolution effect of a true domain wall and the microscope resolution? The authors should make a comment on it.

Reply:

We thank the reviewer for raising this question. For perovskite ferroelectrics, the DW width normally is on the order of a couple of unit cells (~ 1 nm, Refs. 33 and 34). For epitaxial thin films, the strong surface depolarization field can lead to vortex-like chiral rotation of the surface dipole in the vicinity of the DW. Previous TEM studies have shown the lateral extension of this region is on the order of 2-3 nm (Ref. 35). In our study, the spatial (or lateral) resolution of SHG microscope is about 300 nm, which can only probe the diffraction limit of the SHG signal at the DWs, rather than the intrinsic DW width. To clarify this point, we added the SHG microscope resolution to the Method section (first paragraph on page 17) as follows:

“The diffraction limit of the excitation laser beam (spot size) was estimated to be $\lambda/2NA = 380$ nm. Due to the second order nonlinearity of the SHG light, the spatial resolution was estimated to be ~ 300 nm.”

We also added the following sentences to the 1st paragraph on Page 5:

“The width of the enhanced and suppressed SHG signal is about 300-400 nm, which approaches the diffraction limit at this wavelength and the resolution of the SHG microscope (Methods).”

The DW width is also beyond the resolution of the PFM technique, which is about 5 nm for our experimental setup (Ref. 41: Xiao *et al.*, *APL* **103**, 112903 (2013)). In the PFM image shown in Fig. 1c, the DW width is limited by the step size of the line scan (~ 30 nm). The region of $1.5 \mu\text{m}$ width mentioned by the Reviewer is likely the P_{down} domain sandwiched between two P_{up} domains, not the DW, as labeled in Fig. R3a. Figure 1c shows the PFM phase image, where these two polarization states have 180° phase difference.

The DW position is better illustrated in the PFM amplitude image. As shown in Fig. R3b, the P_{up} and P_{down} domains exhibit similar piezoelectric response, and signal diminishes at the DW. We have included this data in the Supplementary Figs. S1c-d, and added the following sentences in the Method section (2nd paragraph on Page 16):

“The resolution of PFM is about ~ 5 nm for our experimental setup,⁴¹ which cannot resolve the intrinsic DW width.”

Fig. R3 | **a**, PFM phase and **b**, amplitude images of the domain structure on PZT shown in Fig. 1c in the main text. The lower panels show the signal profiles along the dotted lines. The blue arrows point to the DW positions. This figure is incorporated as Supplementary Fig. S1c,d.

Comment 1.3:

3. The authors should explain more precisely how to put an MoS₂ film exfoliated from a bulk crystal on PZT. Is it attached on PZT as atomic-scale contact? How is it estimated? Is there no space charges effect or depolarization effect at the interface?

Reply:

Following the reviewer's suggestion, we have expanded Supplementary Information Section 2 (Preparation and Characterization of Monolayer MoS₂ on PbZr_{0.2}Ti_{0.8}O₃). As shown in Fig. R4, we first mechanically exfoliated MoS₂ flakes from a bulk single crystal on a Gel-Film (Fig. R4a), and identified the monolayer flakes via Raman measurements. For the selected flake, its crystalline orientation was identified by the polarized SHG measurement (Fig. R4b). We then created the P_{up} and P_{down} square domains on the PZT film (Figs. R4c-d). As shown in Figs. R4c-f, these PZT films have pre-deposited Au marks (50 μm apart), which can help us locate the domain position and facilitate the alignment of the MoS₂ crystalline axis with the PZT DWs. During transfer, we aligned the a -axis (zigzag orientation) of MoS₂ with the horizontal ([100]) DWs of PZT. After transfer, we used SHG mapping to check if we have achieved the desired alignment (Fig. R4f). This figure is now incorporated in Supplementary Fig. S4.

Fig. R4 | **a**, Optical image of a mechanically exfoliated MoS₂ flake on the Gel-Film. The dashed line indicates the top edge of the 1L piece is along the a -axis. **b**, Polarized SHG measurement reveals the crystalline orientation of the 1L piece shown in **a**. **c**, A 50 nm PZT film patterned with an array of Au marks (50 nm apart). The dotted line highlights the square domains written on this film. **d**, Reflected SHG mapping reveals the DWs of the domain structure. The dashed lines serve as the guide to the eye during the transfer process. **e**, Optical image of the 1L MoS₂ flake transferred on top of the domain structure in PZT. **f**, Reflected SHG mapping of the 1L MoS₂/PZT heterostructure. This figure is incorporated in Supplementary Fig. S4.

We emphasize that the domains on PZT were patterned prior to the transfer of top MoS₂ flake. In this sense, MoS₂ does not serve as an electrical contact as we do not exploit it to switch the polarization underneath. As the domain structure was written in the ambient condition, the freshly polarized charged surface can be well screened by ambient adsorbates (Ref. 26: Song *et al.*, *ACS Appl Mater & Interfaces* **10**, 19218 (2018), Ref. 27: Hong *et al.*, *APL* **97**, 033114 (2010)), which minimizes the depolarization

effect. As MoS₂ is deposited on top of the domain structure via a dry-transfer approach, the charge screening layer on PZT surface should not be affected. Therefore, we do not expect significant depolarization effect in PZT caused by the presence of MoS₂. This conclusion is consistent with the PFM phase image shown in Fig. 1d, where the domain structure is not altered in the presence of MoS₂ top layer. To clarify this point, we added the following sentences in the first paragraph on page 4:

“As shown in Fig. 1d, the presence of the MoS₂ top layer does not alter the domain structure underneath. This is not surprising as the domain structure is written in the ambient condition, where the freshly polarized surface bound charge can be well screened by charged adsorbates.^{26,27} The MoS₂ flake is deposited on top of the domain structure via a dry-transfer approach, which should not affect this surface screening layer on PZT.”

Comment 1.4:

4. The “zigzag axis” (p4-2) and the “armchair direction” (p6-17) are terms in popular parlance. It is preferable to be described with the crystallographic axes.

Reply:

Following the Reviewer’s advice, we added the crystallographic axes of MoS₂ and PZT in the insets of figures, similar as those shown in Figs. R1-R4, and revised the sentences in the 1st paragraph on Page 4 as follows:

“During transfer, the *a*-axis of MoS₂ (zigzag orientation) was aligned with the horizontal DWs ([100] direction in PZT) (see Methods for transfer details).”

Comment 1.5:

5. What do the authors want to conclude from the result of PL spectra (Fig. 1e and f). Give briefly concrete and specific facts related to this hetero-system.

Reply:

By fitting to the PL spectra in Fig. 1e (Supplementary Fig. S5), we find that the ferroelectric polarization can effectively modulate the PL intensity associated with the neutral exciton and negative trions (A⁻) in monolayer MoS₂. Such modulation of PL has previously been reported in WS₂ (Ref. 23: Li *et al.*, *Omega* **1**, 1075 (2016)), MoSe₂, and WSe₂ (Ref. 24: Wen *et al.*, *ACS Nano* **13**, 5335 (2019)), and has been attributed to the ferroelectric polarization induced doping effect, which modulates the carrier density in MoS₂. The fact that we observe the doping induced PL modulation confirms that our MoS₂ sample is in close contact with the PZT surface. To clarify this point, we added the following sentence to the 2nd paragraph on Page 4:

“Such modulation of PL spectra in TMDCs via neighboring ferroelectric domains has previously been attributed to the polarization induced doping effect,^{23,24} and confirms the close interfacial contact between MoS₂ and PZT in our samples.”

Furthermore, the reflected SHG signal does not show appreciable contrast on the P_{up} and P_{down} domains, suggesting that it is not affected by the interfacial charge coupling. To clarify this point, we added the following sentence to the last paragraph on Page 5:

“Unlike the PL data (Figs. 1e-f), no prominent difference in the SHG signal has been observed in the regions on the P_{up} and P_{down} domains, confirming the signal is not affected by the interfacial charge coupling between MoS₂ and PZT.”

We also note that the relative strength of the PL modulation can vary due to the existence of charge screening layer on the PZT surface, which depends sensitively on the sample preparation details, such as the time interval between domain patterning and TMDC transfer, the transfer temperature, speed, and environment (dry vs. wet), *etc.* We added this discussion in the Supplementary Information (Section 2).

Comment 1.6:

6. Put the orientation of domain walls of PZT on Fig. 1c. The thicknesses of MoS₂ and PZT should be also given in the early part of the manuscript.

Reply:

We have followed the reviewer’s suggestion, and made the following revisions to the main text.

1. We labeled the crystalline orientation of the DWs in all relevant figures as insets.
2. We modified the following sentences on Pages 3:

“For the ferroelectric layer, we worked with 20-50 nm thick epitaxial PZT thin films deposited on (001) SrTiO₃ substrates, with La_{0.67}Sr_{0.33}MnO₃ (LSMO) (10 nm) buffer layers serving as the bottom electrode (Methods). The PZT films are (001)-oriented with out-of-plane polar axis (Supplementary Fig. S1). Selected 1L MoS₂ flakes were transferred on top of the PZT thin film above a region patterned with a series of square domains with alternating up (P_{up} or [001]) and down (P_{down} or [00 $\bar{1}$]) polarization. ... where the horizontal (vertical) DWs are along the [100] ([010]) orientation of PZT.”

Response to Reviewer #2's Comments

We thank Reviewer 2 for considering the observed nonlinear optical filtering effect as “unconventional” and “interesting”. The reviewer has raised two major questions/concerns regarding: 1) the validity of the proposed model to account for the in-plane polarization at the ferroelectric domain wall, and 2) the significance and potential technological impact of the work. To address the reviewer’s questions, we have expanded the discussion section in the main text, and included additional references. Below is the point-by-point response to the reviewer’s comments.

Comment 2.1:

1. The assumed polarization configuration in Fig. 3a may not be correct. Various experiments had proven that the 180° domain walls are very sharp (within one unit cell in most cases), the Science article by Jia et al about the continuous polarization rotation in a 180° domain wall might be misleading, since it involves elastic deformation in the domain wall region, making it a higher energy configuration. Hence, such continuous polarization rotation should not occur often in the system. The actual polarization configuration created by the PFM tip should follow the electric field lines generated by the electrode configuration, which could be simulated using finite element software, such as ANSYS, COMSOL, etc.

Reply:

We thank the reviewer for raising this important point. First, as the reviewer noted, the 180° DW in bulk PZT is very sharp due to the high energy penalty associated with polar rotation (Ref. 33: Meyer & Vanderbilt, *PRB* **65**, 104111 (2002), Ref. 34: Hlinka & Marton, *PRB* **74**, 104104 (2006)), which can result in charged DW. This condition, however, does not apply to the surface state of epitaxial thin films, where tilting of local dipoles can mitigate the strong depolarization field and thus compensates this energy cost. For example, previous studies have shown that up to 6 unit cells of ferroelectric surface layer have reduced polarization (Ref. 36: Gao et al., *Nat. Commun.* **7**, 11318 (2016)). Besides the continuous polar rotation observed at the surface of 180° DWs (Ref. 35: Jia et al., *Science* **331**, 1420 (2011)), the presence of extremely strong depolarization field can even induce polar vortices, as in the case of PbTiO₃/SrTiO₃ superlattices (e.g., Ref. 6: Yadav et al., *Nature* **565**, 468 (2019), Ref. 7: Zubko et al. *Nature* **534**, 524 (2016)), where theoretical modeling has shown that the polar anomaly is dominated by the interface contribution (Ref. 7). To clarify this point, we added the following discussion to the first paragraph on Page 8:

“In fact, the 180° DW in bulk PZT is on the order of a couple of unit cells and does not acquire an in-plane component.^{33,34} The chiral rotation of the local dipole, which is critical for forming the in-plane polarization, can only be stabilized at the surfaces/interfaces (Fig. 3a) due to the presence of strong depolarization field.^{35,36} For example, previous transmission electron microscopy (TEM) studies have revealed a flux-closure polar structure at the surface of the DW in PZT thin films,³⁵ and even emergence of polar vortices in PbTiO₃/SrTiO₃ superlattices,^{6,7} where theoretical modeling has pointed to the dominant role of the interface contribution to the polar anomaly.⁷”

Second, following the reviewer’s suggestion, we have carried out finite element analysis (Ansoft Maxwell 3D V.14) to simulate the electrical potential and field distribution in PZT upon applying a bias voltage (V_{bias}) to a point contact (conductive AFM tip), with the result shown in Figs. R5a-b. Figure R5c shows a schematic view of the local dipole distribution at a 180° DW based on the scenario proposed by

the Reviewer. The underlying assumptions are: 1) the local dipole orientation follows the electric field line direction while V_{bias} is applied, and 2) it is frozen to the same orientation after V_{bias} is removed, *i.e.*, the AFM probe is scanned away from the location. Here we assume the positively biased conductive AFM probe scans from 0 to $+x$ direction, switching a P_{up} domain into P_{down} domain. As a result, the polar re-orientation only occurs at $-x$ region, as the $+x$ region will be written repetitively during the subsequent AFM scan.

Fig. R5 | **a**, Finite element analysis of the electrical potential distribution in a 50 nm PZT film with global 10 nm LSMO bottom electrode at AFM tip bias of 1 V applied to a point contact (AFM tip), with **b**, the expanded view close to the top surface of PZT. The arrows show the simulated local electric field line direction. **c**, Schematic view of a positively biased scanning AFM tip switching a P_{up} domain to P_{down} domain, assuming the local dipole follows the electric field line direction. The blue (red) arrows mark the dipole direction within a domain (DW). The dotted lines indicate the electric field lines when the biased AFM tip locates at $x = 0$. The dashed lines mark the boundaries of the DW. This figure is incorporated as Supplementary Fig. S10.

Based on this picture, the lateral polarization P_{\parallel} can persist to the bulk region of the film (well exceeds 5 nm), and it points from the P_{down} domain to the P_{up} domain. This is in sharp contrast to the flux-closure model (Fig. 3a), where the chiral dipole rotation is confined within the surface 2-3 nm, and the overall P_{\parallel} points from the P_{up} domain to the P_{down} domain. We believe the flux-closure type DW is energetically more favorable for two reasons.

1. In the bulk region of PZT film, the lateral dipole orientation cannot be sustained at the DW after V_{bias} is removed, as it significantly increases both the electrostatic energy (leading to charged DW) and elastic energy. This energy anisotropy between different crystalline axes is especially strong in PZT films on STO due to the large compressive (3.6%) strain.
2. At PZT surface, this model depicts an abrupt change of polar angle, *i.e.*, close to 90° polar angle change, at the interface with the P_{up} domain (Fig. R5c), which closely resembles the emergence of a domain in a c -axis oriented film. The 90° DW is charged and naturally wider than the 180° DW. It has been shown theoretically that widths for the 90° and 180° DW are 3.6 nm and 0.6 nm, respectively (Ref. 34: Hlinka and Marton, *PRB* **74**, 104104 (2006)). This means the spatial extension of this kind of DW configuration is comparable with that of the flux-closure type DW (2-3 nm), while the 90° polarization change corresponds to a much higher electrostatic energy than the gradual polar rotation.

We thus conclude that the chiral dipole rotation model is more favorable in our systems. To clarify this point, we included the related discussion in the Supplementary Information Section 4, and added Fig. R5 as Fig. S10. We also revised the following sentences in the 2nd paragraph on Page 8:

“To examine the feasibility of the interfacial polar coupling scenario, we exploited a phenomenological model to estimate the net in-plane polarization at the DW based on the TEM study in Ref. [35], considering a triangle-shaped flux-closure domain structure that hosts continuous electric dipole rotations at the surface of a 180° DW (Fig. 3a). Compared with an a -domain like DW configuration induced by the local electric field of a biased AFM tip (Supplementary Fig. S10), this model depicts a dipole distribution with comparable (if not smaller) spatial extension but lower electrostatic energy.³⁴”

Comment 2.2:

2. Since the purpose of this work is to prove polarization coupling between the PZT film and monolayer MoS₂, the experiments could be performed more conveniently if one could just using interdigital electrode on one surface of the film, then pole the thin film into a periodic alternating in-plane polarization pattern. One could control the in-plane polarization strength by applying different level of electric field. It would be better quantified and more convincing than the first principles estimate for the in-plane polarization of a domain wall, since the first principles calculations could not give accurate quantitative results. Why introduce domain walls into the picture if one just wants to make the polarization of monolayer MoS₂ to couple with an in-plane polarization field?

Reply:

With respect, we'd like point out that the purpose of this work is *not* to prove the polarization coupling between PZT and MoS₂. The key novelty of this work is to demonstrate a simple yet effective material scheme to enable reconfigurable tailoring/filtering of the SHG signal at the nanoscale, as highlighted in the title of the manuscript. To realize such filtering effect, we take advantage of two facts: 1) Ferroelectric domain wall can be created in a controlled fashion with nanoscale precision via the scanning probe approach in single crystalline ferroelectric thin films. It is worth noting that scanning probe writing of local domain structure can be realized even in the presence of MoS₂ top layer, as demonstrated in Ref. [22] (Lipatov *et al.*, *Nano Lett.* **19**, 3194 (2019)), as MoS₂ is a wide bandgap semiconductor. 2) The lateral polarization at the 180° DW surface is comparable with that of MoS₂, so the tuning of the SHG signal can be highly efficient. The method proposed by the Reviewer, although interesting in its own right, is not suited to our purpose. It is much more complicated as it involves lithographically fabricated electrodes, and is not reconfigurable once the electrodes have been fabricated.

Also, we exploited the first-principles modeling to have a ballpark estimation of the polarization of these two material systems, which is sufficient to confirm the proposed model is realistic. As the polarization value does not directly translate to the SHG signal, a precise matching of the polarization value is not critical. Proper modeling of the angle-dependent SHG response has been carried out via the nonlinear optical susceptibility tensor (d -tensor) calculation, and can give an excellent description of the experimentally observed tailoring/filtering patterns (Fig. 3f). To clarify these points, we added the details of the d -tensor calculation on Pages 10-12 and the Supplementary Section 5, and added the following sentence to the last paragraph on Page 9 as follows:

“While the precise value of the polarization may vary, this simple model naturally explains the major features of our observation.”

Comment 2.3:

3. When the monolayer MoS₂ is deposited on a substrate with existing polarization pattern, the polarization of the two would either enhance or cancel each other, which is nothing special about. Although the experimental procedure is well executed in this work, there is no new concept generated and I am also having a hard time to see the purpose of this research effort. Obviously, it would be a poor optical filter since the PZT film is not a transparent material. There are many other ways to make better optical filters, the authors need show the advantages of such filters over existing filters in order to let the readers see the value of this research.

Reply:

First, we'd like to emphasize that the observed SHG filtering phenomenon is not due to the direct addition of the polarization field. MoS₂ is non-centrosymmetric, but it does *not* have a net polarization. It has a three-fold rotational symmetry, with three equivalent polar axes (120° apart). If one simply superimposes all polarization fields at the MoS₂/PZT interface, only the PZT term would survive. This is clearly not what we observe. As a matter of fact, polarization is a Berry phase effect intrinsic to each material (Supplementary Ref. 16: King-Smith & Vanderbilt, Theory of polarization of crystalline solids, *PRB* **47**, 1651 (1993)), and the net polarization of the composite system cannot be directly superimposed. We realized that this confusion arises from our previous discussion of the SHG response in the transmitted mode, where we defined an intermediate variable $\vec{P}_{interface} = \vec{P}_{MoS_2} + \vec{P}_{out}$. Our initial intention was to depict the rotation of the noncentrosymmetric axis that modifies the propagation direction of the E&M wave. This term, however, is misleading and not a meaningful physical quantity. We thank the reviewer for bringing up this important point, and removed the discussion associated with $\vec{P}_{interface}$ as well as the schematic in original Fig. 4e. We also revised the last paragraph on Page 9 as follows:

“When one of the polar axes of MoS₂ is aligned with the in-plane polarization of PZT at the DW regions, as for the odd DWs, their excited interfacial SH dipole fields are coherently coupled,^{37,38} leading to significantly enhanced SHG response that is linearly polarized along the polar axis.”

Second, we'd like to bring to the Reviewer's attention that the tailored SHG pattern not only enhances or cancels the SHG signal originating from the polar axis of MoS₂ that's aligned with the DW polarization, but also filters the SHG signal coming from the other two polar axes that are rotated by 120°. As shown in Fig. 2b, such filtering effect is conventionally achieved by applying an optical analyzer between the sample and the detector. With the presence of the PZT DW, efficient filtering has been achieved without the need of an analyzer (Fig. 2e). In another word, we have already demonstrated that the DW of PZT can replace an existing optical filter to achieve highly efficient filtering of the light polarization of the SHG signal. Compared to an optical analyzer, the MoS₂/PZT composite structure has the distinct advantages in terms of size scaling, offering the opportunities to achieve on-chip generation and filtering of SHG signals. The fact that it is possible to create nanoscale domain structures in PZT through a MoS₂ top layer (Ref. 22) can further lead to reconfigurable functionalities in nano-optics. These are in sharp contrast to the existing optical filter technologies, which are macroscopic in terms of dimensions, time consuming in terms of optical setup, and cannot be programmed at the nanoscale. To highlight these points, we revised the last paragraph on Page 12 as follows:

“Comparing the results shown in Fig. 2b and Fig. 2e, it is clear that the lateral polarization of PZT DW can replace an optical analyzer to provide efficient filtering of the light polarization for the SHG

signal of MoS₂. It further enhances or quenches the SHG intensity for the selected light polarization depending on the underlying DW chirality. Compared with the existing optical filter technologies, which are macroscopic in terms of dimensions, time consuming in terms of optical setup, and cannot be programmed at the nanoscale, the MoS₂/PZT heterostructure has the distinct advantages in terms of size scaling and being nanoscale reconfigurable, offering the opportunities to achieve on-chip generation and smart filtering of SHG signals for nano-optics.”

Third, we’d like to clarify that the DW filtering effect occurs at the reflection mode, so whether PZT is transparent or not is not relevant. Also, in the revised Fig. 4f, we compared the reflected and transmitted SHG signal in MoS₂/PZT heterostructures with different PZT thicknesses. As we are working with very thin PZT films, the SHG intensity for transmission mode can be comparable with that of the reflection mode within certain thickness range.

Comment 2.4:

4. “The SHG signal is solely mediated by the interfacial polar symmetry” is not an accurate statement. In my opinion, the SHG is still mediated by the combined polarization of the film and the monolayer MoS₂, not by the symmetry. Symmetry does not “couple”.

Reply:

With respect, we disagree with the Reviewer and believe the statement of “The SHG signal is solely mediated by the interfacial polar symmetry” is accurate. Symmetry as an operation can be coupled. For example, MoS₂ belongs to the point group D_{3h}, which possesses a mirror plane perpendicular to the principle axis (σ_h). Coupling to the vertical polarization of PZT at the interface removes this mirror symmetry, reducing it to point group C_{3v}. Similarly, coupling with the parallel polarization at the PZT DW further removes its 3-fold rotational symmetry, reducing it to point group C_s.

As discussed in the response to Comment 2.3, the observed tailoring effect is not originating from the direct superposition of polarization fields from MoS₂ and PZT. Instead, the symmetry coupling of the MoS₂/PZT system is captured in the composite second order d -tensor $d_{interface}^{(2)}$, which is the superposition of the d -tensors of the individual constituent layers $d_{interface}^{(2)} = d_{MoS_2}^{(2)} + d_{DW}^{(2)}$. To clarify this point, we added the modeling details for the reflected SHG signal to Pages 10-12 in the main text and Supplementary Section 5. As shown in Fig. 3f and Supplementary Table S2, this symmetry coupling picture can give an excellent description of the experimental results shown in Figs. 2e and 3e.

In addition, we introduced this concept in the context of a broader discussion of interfacial interactions between functional materials, including charge (*e.g.*, charge transfer, electric field effect), spin (*e.g.*, magnetoelectric coupling, proximity effect), lattice (*e.g.*, epitaxial strain), and symmetry (*e.g.*, topology, superlattice formation, magnetic anisotropy) related mechanisms. These mechanisms are often entangled and not entirely independent. The most well known example of symmetry-driven effect is the Moiré pattern between two van der Waals materials, *e.g.*, graphene/h-BN and twisted bilayer graphene, which has led to the discovery of a wide range of exotic many body effects (Refs. [8-15]). Our work represents the *first* study leveraging the symmetry coupling between complex oxides and van der Waals materials. Similar to the other three terms (charge, spin, and lattice), the term “symmetry” is exploited to capture the core feature that drives the observed interfacial synergy effect and cannot be taken out of context.

Response to Reviewer #3's Comments

We thank the Reviewer for the highly positive comments on the novelty and potential scientific and technological interest of this work. The reviewer also commended on the validity of “*the SHG results in reflection and of their interpretation*”. The reviewer, however, has raised questions about the experimental and modeling details, as well as the interpretation of the observed SHG tuning effects in the transmission mode. We thank the reviewer for the critical comments and valuable suggestions, which help us improve the technical strength of the work. To address these comments, we carried out additional experiments on new PZT/MoS₂ samples and included more in-depth SHG modeling, as well as making substantial revisions to the text, with the details shown below.

Comment 3.1:

3. VALIDITY

a) The work is convincing in terms of the SHG results in reflection and of their interpretation, although it would be suitable to complement them with a more comprehensive SHG analysis.

Reply:

In the revised version, we added a detailed discussion of the *d*-tensor modeling of the stacking angle-dependence of the reflected SHG response in the manuscript and Supplementary Information Section 5, as will be discussed in detail in the response to Comment 3.3.

Comment 3.2:

*b) The argumentations provided for the SHG results obtained in transmission (Fig. 4), including statements on the coupling of the out-of-plane polarization of PZT to the polar axis of MoS₂ (e.g. line 17 on page 9) appear less convincing and need thorough revision. The discussion provided on pages 9-10 has also a number of flaws, such as: 1) the refractive index used for PZT ($n=10$) is clearly wrong: it should be ~ 2.4 (the authors here incorrectly use the value of the dielectric constant at low frequencies to extrapolate the refractive index at optical frequencies); 2) the focusing action of the PZT layer cannot be used to justify a better collection efficiency, since Snell's law applies also for the bottom interface (PZT/air), unless a different collection scheme is used?; 3) for transverse EM waves propagating along the *z*-axis, all SH contributions from the *z*-polarization (P_{out}/P_{in}) should be zero in the far field.*

Reply:

We thank the reviewer for bringing up these important points. 1) The reviewer is correct that we used the wrong refractive index for PZT.

2) We fully agree with the reviewer that the vertical polarization ($\pm z$) of PZT cannot be used to explain the observed tailoring effect in the transmitted SHG signals, and have removed the original Fig. 4e and the related discussion from the main text.

3) As the transmitted SHG signal show clear contrast between the P_{up} and P_{down} domains, we have considered alternative scenarios that can contribute to this difference. In our current experiments, we have set the focus plane of the detector at the MoS₂/PZT interface ($z = 0$). To identify the relevant dielectric layer that's responsible for the tailoring effect, we captured a series of SHG images in the de-focused

condition by gradually increasing the sample-detector distance (lowering the objective position in $-z$ direction). As shown in Fig. R6, the SHG signal gradually diminishes during de-focusing. The contrast between the P_{up} and P_{down} domains are still very sharp at $z = -2 \mu\text{m}$, and becomes unrecognizable at $z = -14 \mu\text{m}$. Considering the thicknesses of PZT (50 nm) and the $\text{La}_{0.67}\text{Sr}_{0.33}\text{MnO}_3$ (10 nm) buffered STO substrate (0.5 mm), the relevant intensity of the collected transmission signal is fully attenuated in STO before reaching air. We thus conclude that the tailoring effect occurs at the PZT layer. To highlight this point, we included Fig. R6 and the related discussion in the Supplementary Information (Section 3 and Fig. S8), and revised the 1st paragraph on Page 14 as follows:

“We next investigated the effect of focus plane on the transmission image, which shows that SHG signal is fully attenuated in the STO substrate (Supplementary Fig. S8), confirming that the relevant dielectric layer for this tuning effect is indeed PZT.”

Fig. R6 | Transmitted SHG images of a $\text{MoS}_2/50 \text{ nm}$ PZT heterostructure at different sample-detector distances. The objective is first focused at the MoS_2/PZT interface with **a**, $z = 0$, and then **b-h**, gradually moved away from the sample by lowering the objective positions (decreasing z). The dashed lines serve as the guide to the eye. This figure is included as Supplementary Fig. S8.

To further investigate the role of PZT, we have carried out the SHG measurements on MoS_2/PZT heterostructures with different PZT layer thickness. Figure R7 compares the SHG results obtained on heterostructures with 20 nm, 30 nm, and 50 nm thick PZT films. Despite the different PZT thicknesses, all samples exhibit qualitatively similar reflected SHG responses, with alternating enhancement and suppression of the SHG signal at the horizontal DWs ([100]) and unappreciable SHG contrast at the vertical ([010]) DWs. To quantitatively examine this effect, we measured the SHG intensity on the P_{up} and P_{down} domains as a function of film thickness. As shown in the Fig. R7d, the reflected SHG signal has similar intensity in both polarization states, which does not show appreciable dependence on the film thickness. This result further confirms that the reflected SHG tuning pattern is independent of PZT’s thickness, yielding additional support to its interfacial nature. The transmitted signals also show qualitatively similar contrast between the P_{up} and P_{down} domains. The SHG intensity, however, increases monotonically with film thickness in both polarization states, suggesting that the signal has a bulk origin.

It also explains why the signal at the DWs, which is an interfacial effect, is not appreciable in the transmitted image.

Fig. R7 | **a-c**, Reflected (upper) and transmitted (lower) SHG images of 1L MoS₂ on **a**, 20 nm, **b**, 30 nm, and **c**, 50 nm PZT films patterned with square P_{up} and P_{down} domains, with the crystallographic orientations of MoS₂ and PZT indicated as insets. The scale bars are 3 μ m. The dashed lines highlight the DW positions. All images were taken at the excitation laser power of 20 mW. These figures are incorporated in Supplementary Fig. S7. **d**, Average SHG intensity as a function of PZT thickness taken on the P_{up} (squares) and P_{down} (triangles) domains in both reflected (open symbols) and transmitted (solid symbols) modes. This figure is incorporated as Fig. 4f.

To highlight this point, we included Fig. R7a-c as Supplementary Fig. S7, Fig. R7d as Fig. 4f, and added the following discussion to 1st paragraph on Page 14:

“To understand the origin for this ferroelectric polarization-dependent SHG response, we quantitatively compared the signal intensity in 1L MoS₂/PZT heterostructures with different PZT layer thicknesses for both detection modes (Fig. S7). As shown in Fig. 4e, the intensity of the reflected SHG signal does not exhibit apparent dependence on PZT thickness, consistent with its interfacial origin. The transmitted light intensity, on the other hand, increases monotonically with the layer thickness of PZT for both P_{up} and P_{down} domains, with the signal at the P_{up} domain approaching the intensity for the reflected signal in the heterostructure with 50 nm PZT, which suggests that this tailoring effect is related to the bulk state of PZT.”

We also revised the concluding paragraph on Page 15 as follows:

“The transmitted SHG signal, in sharp contrast, is sensitively tuned by the out-of-plane ferroelectric polarization rather than the DW, which is attributed to the bulk state of PZT.”

A possible scenario is that the PZT layer serves as an optical cavity, which can lead to enhanced SHG signal at certain layer thickness, *i.e.*, through constructive interference among multiple reflections. Confirming this scenario, however, requires working with a wide range of PZT layer thickness, ideally larger than half of the optical wavelength ($\lambda_{air}/2n_{PZT} \approx 83$ nm). This is beyond our current experimental capacity, as the PZT film much thicker than 50 nm is no longer fully strained. The strain is relaxed through the growth of *a*-domain, which also possesses in-plane polarization and can generate SHG signal (Ref. 32: De Luca *et al.*, *Adv Mater.* **29**, 1605145 (2017)). On the other hand, as the tailoring

effect in transmission mode is originating from the bulk state of PZT rather than the MoS₂/PZT interface, identifying its precise mechanism is also beyond the scope of the current work, which focuses on the interfacial polar coupling between MoS₂ and PZT. We included this discussion in the 1st paragraph on Page 14 as follows:

“The observed film thickness dependence is opposite to what’s expected due to light absorption in a non-transparent dielectric layer. We thus speculate that the tailoring of the transmitted SHG originates from a possible cavity effect of the PZT through constructive interference among multiple reflections. To verify this scenario, however, requires working with much thicker PZT films, ideally larger than half wavelength of the SHG light ($\lambda_{air}/2n_{PZT} \approx 83 \text{ nm}$, with $n_{PZT} \approx 2.4$). This is challenging as the PZT films thicker than 50 nm tend to relax the epitaxial strain through forming *a*-domains,³² making the local polarization orientation not well defined.”

Comment 3.3:

4. DATA, METHODOLOGY & STATISTICAL ANALYSES

*a) It would be appropriate to include a more thorough analysis of the results of Fig. 2-3 in terms of SHG theory and composition of the *d*-matrices, which is at the moment lacking in the main text and only briefly touched upon in Suppl. Information.*

Reply:

Following the reviewer’s suggestion, we have added a detailed discussion of the *d*-tensor modeling of the reflected SHG response on Pages 10-12 as follows:

“The net SHG response for each of these DWs can be well modeled using the nonlinear electromagnetic theory, considering the second-order nonlinear optical susceptibility tensors for the MoS₂/PZT heterointerface. As the thickness of MoS₂ and the depth of the flux-closure region at PZT DW (*h*) are well below the optical wavelength, the susceptibility tensor (or the contracted *d*-tensor) of the composite system equals to the sum of the adjacent layers: $d_{interface}^{(2)} = d_{MoS_2}^{(2)} + d_{DW}^{(2)}$, where $d_{MoS_2}^{(2)}$ and $d_{DW}^{(2)}$ are the *d*-tensors for MoS₂ and DW, respectively. For 1L MoS₂ with *D*_{3h} point group, the second-order *d*-tensor can be expressed as:²⁹

$$d_{MoS_2}^{(2)} = \begin{pmatrix} 0 & 0 & 0 & 0 & 0 & d'_{16} \\ d'_{21} & d'_{22} & 0 & 0 & 0 & 0 \\ 0 & 0 & 0 & 0 & 0 & 0 \end{pmatrix}, \quad (3)$$

where $d'_{21} = d'_{16} = -d'_{22} = d_{MoS_2}$. For the tetragonal PZT thin films with 4mm point group symmetry, the *d*-tensor can be written as:³⁹

$$d_{PZT}^{(2)} = \begin{pmatrix} 0 & 0 & 0 & 0 & d_{15} & 0 \\ 0 & 0 & 0 & d_{15} & 0 & 0 \\ d_{31} & d_{31} & d_{33} & 0 & 0 & 0 \end{pmatrix}, \quad (4)$$

where $d_{15} = d_{31}$, and $d_{33} \approx 0.9d_{15}$. The tensor elements for PZT were obtained by averaging the calculated and experimental values.^{31,39} In our work, the crystallographic axes of PZT coincide with the experimental reference frame (*x*, *y*, *z*), where the [001] orientation of PZT is along the *z*-axis (Fig. 1a). We first considered a square *P*_{down} domain embed in a *P*_{up} region in PZT (Fig. 3c), with the

horizontal ([100]) DW aligned with the a -axis of MoS₂ (stacking angle $\theta = 0^\circ$). The interfacial composite tensors at the P_{up} and P_{down} domains are given by:

$$\begin{aligned} d_{P_{\text{up}}}^{\text{interface}} &= \begin{pmatrix} 0 & 0 & 0 & 0 & d_{15} & d_{\text{MoS}_2} \\ d_{\text{MoS}_2} & -d_{\text{MoS}_2} & 0 & d_{15} & 0 & 0 \\ d_{15} & d_{15} & d_{33} & 0 & 0 & 0 \end{pmatrix}, \\ d_{P_{\text{down}}}^{\text{interface}} &= \begin{pmatrix} 0 & 0 & 0 & 0 & -d_{15} & d_{\text{MoS}_2} \\ d_{\text{MoS}_2} & -d_{\text{MoS}_2} & 0 & -d_{15} & 0 & 0 \\ -d_{15} & -d_{15} & -d_{33} & 0 & 0 & 0 \end{pmatrix}. \end{aligned} \quad (5)$$

As the lateral polarization for the flux-closure domain is comparable with the bulk polarization of PZT, we obtained the d -tensors for the four DWs (Top-DW, Bottom-DW, Left-DW, and Right-DW) via a rotation matrix transformation (Supplementary Information):³¹

$$\begin{aligned} d_{\text{Top-DW}}^{(2)} &= -d_{\text{Bottom-DW}}^{(2)} = \begin{pmatrix} 0 & 0 & 0 & 0 & 0 & d_{15} \\ d_{15} & d_{33} & d_{15} & 0 & 0 & 0 \\ 0 & 0 & 0 & d_{15} & 0 & 0 \end{pmatrix}, \\ d_{\text{Left-DW}}^{(2)} &= -d_{\text{Right-DW}}^{(2)} = \begin{pmatrix} d_{33} & d_{15} & d_{15} & 0 & 0 & 0 \\ 0 & 0 & 0 & 0 & 0 & d_{15} \\ 0 & 0 & 0 & 0 & d_{15} & 0 \end{pmatrix}. \end{aligned} \quad (6)$$

The interfacial composite tensors at the DWs can thus be expressed as:

$$\begin{aligned} d_{\text{Top-DW}}^{\text{interface}} &= \begin{pmatrix} 0 & 0 & 0 & 0 & 0 & d_{\text{MoS}_2} + d_{15} \\ d_{\text{MoS}_2} + d_{15} & -d_{\text{MoS}_2} + d_{33} & d_{15} & 0 & 0 & 0 \\ 0 & 0 & 0 & 0 & d_{15} & 0 \end{pmatrix}, \\ d_{\text{Bottom-DW}}^{\text{interface}} &= \begin{pmatrix} 0 & 0 & 0 & 0 & 0 & d_{\text{MoS}_2} - d_{15} \\ d_{\text{MoS}_2} - d_{15} & -d_{\text{MoS}_2} - d_{33} & -d_{15} & 0 & 0 & 0 \\ 0 & 0 & 0 & 0 & -d_{15} & 0 \end{pmatrix}, \\ d_{\text{Left-DW}}^{\text{interface}} &= \begin{pmatrix} d_{33} & d_{15} & d_{15} & 0 & 0 & d_{\text{MoS}_2} \\ d_{\text{MoS}_2} & -d_{\text{MoS}_2} & 0 & 0 & 0 & d_{15} \\ 0 & 0 & 0 & 0 & d_{15} & 0 \end{pmatrix}, \\ d_{\text{Right-DW}}^{\text{interface}} &= \begin{pmatrix} -d_{33} & -d_{15} & -d_{15} & 0 & 0 & d_{\text{MoS}_2} \\ d_{\text{MoS}_2} & -d_{\text{MoS}_2} & 0 & 0 & 0 & -d_{15} \\ 0 & 0 & 0 & 0 & -d_{15} & 0 \end{pmatrix}. \end{aligned} \quad (7)$$

Furthermore, we derived the explicit expressions of interfacial SHG tensors at the four DWs as a function of stacking angle θ , which are given in the Supplementary Information (Eq. S5). The SH dipole field $\mathbf{P}_{\text{interface}}^{2\omega}$ is given by the product of d -tensors and the fundamental field, and the SHG intensity at the 1L MoS₂/PZT interface is given by:

$$I_{\text{SHG}}(\varphi = 0^\circ) \approx |\mathbf{P}_{\text{interface}}^{2\omega}(\varphi = 0^\circ)|^2. \quad (8)$$

... Figure 3f shows the simulated SHG results, which capture well the features of SHG tailoring effect for all stacking angles (Figs. 2e and 3e). Supplementary Table S2 lists a detailed comparison between the experimental and modeling results, which shows an excellent agreement, yielding strong support to the scenario for the interfacial polar coupling between MoS₂ and PZT DW.”

We also included the full modeling details, the explicit expressions of interfacial SHG tensors at the four DWs as a function of stacking angle θ , in the Supplementary Information, Section 5.

Comment 3.4:

b) The authors should provide more detailed information on the SHG experimental setup and measurement conditions (e.g. spot size and focusing), in addition to what is currently provided on p. 12. This will also be useful to understand the SHG transmission results.

Reply:

We added detailed information of the experimental setup for the SHG measurement and the focusing condition for the fundamental beam to the Method section (Pages 16-17) as follows:

“The laser beam passed a polarizer with normal incidence, and then was guided by mirrors into a laser scanning microscope (LSM). ... The sample was placed on a glass slide (1 mm), lying in the x - y plane, which is placed above the 1” diameter stage opening. The incident light was transversely polarized and directed to the sample surface along $-z$ direction, and the excited SHG signals were collected in both reflection ($+z$) and transmission ($-z$) geometries by photomultiplier detectors (PMTs). ... In the LSM, the transmission and reflection modes share the same focus plane using the same type of objective lens (NA1.05/water immersed/WD2.0 mm, 25 \times), so both measurements can be performed simultaneously. The focusing to the MoS₂/PZT interface was performed through the reflected mode. The signal was first collected with no analyzer inserted, and then with an analyzer inserted in different angles with respect to the polarizer orientation. ... The diffraction limit of the excitation laser beam (spot size) was estimated to be $\lambda/2NA = 380$ nm. Due to the second order nonlinearity of the SHG light, the spatial resolution was estimated to be ~ 300 nm.”

Comment 3.5:

c) As done for PZT (Fig. S3), it would be good to provide measurements for SHG on MoS₂-alone and MoS₂ on $\pm z$ PZT (as references), both for transmission and reflection. Current information provided by Fig.S5 is limited (and not very clear) in this respect.

Reply:

Following the reviewer’s suggestion, we carried out SHG measurements on a MoS₂ sample (Fig. R8) as it was exfoliated on Gel-Film and after it was transferred on a PZT domain. As Gel-Film is a soft material, the SHG measurements were taken at a significantly lower laser power to avoid sample damage before transfer. For MoS₂ on Gel-Film, the sample exhibits similar SHG response in the reflection and transmission modes (Figs. R8c-d). For MoS₂ on PZT, we observed distinct SHG patterns in these two detection modes (Figs. R8g-h), similar to those shown in Figs. 2e and 4a. In the reflection mode, the P_{up} and P_{down} domains show comparable SHG response, while the DWs show a tailored pattern. In contrast, in the transmission mode, the P_{up} and P_{down} domains show distinct SHG intensities, while the signal at the DWs does not show appreciable contrast. We included this data in Supplementary Fig. S4, and revised the following sentences in the 1st paragraph on Page 5:

“For both MoS₂ on Gel-Films and bare PZT, the SHG signals detected in the transmission mode exhibit qualitatively similar behavior as in the reflection mode (Supplementary Fig. S3 and Fig. S4c-d).”

Fig. R8 | **a**, Optical image of 1L MoS₂ flake on Gel-Film, **b**, with the polarized SHG measurement revealing its crystalline orientation (dashed line in **a** indicating the zigzag orientation). **c-d**, The corresponding SHG images taken in **c**, reflection and **d**, transmission modes. **e**, SHG mapping of a 50 nm PZT film patterned with an array of Au marks (50 nm apart) reveal the DWs of the domain structure. The polarizations of the domain structure are labeled in Fig. R6a. The dashed lines serve as the guide to the eye during the transfer process. **f**, Optical image of the 1L MoS₂ flake transferred on top of the domain structure in PZT, with the corresponding SHG images taken in **g**, reflection and **h**, transmission modes. The red solid arrows indicate the incident light polarization. No analyzer is applied in the SHG measurements. The scale bars are 10 μm. This figure is incorporated as Supplementary Fig. S4.

Comment 3.6:

d) A further comment on the procedure used to align the zig-zag axis of the MoS₂ flake to the PZT DW and an estimate of the associated uncertainty on the stacking angle (theta) for all shown data, including theory-experiment comparisons in table S2, should be included.

Reply:

Following the reviewer's suggestion, we expanded Section 2 (Preparation and Characterization of Monolayer MoS₂ on PbZr_{0.2}Ti_{0.8}O₃) in the Supplementary Information. As shown in Fig. R8a-b, we first mechanically exfoliated MoS₂ flakes from a bulk single crystal on a Gel-Film (Fig. R8a), and identified the monolayer flakes via Raman measurements. For the selected flake, its crystalline orientation was identified by the polarized SHG measurement (Fig. R8b). We then created the P_{up} and P_{down} square domains on the PZT film. As shown in Figs. R8e-f, these PZT films have pre-deposited Au marks (50 nm apart), which can help us locate the domain position and facilitate the alignment of the MoS₂ crystalline axis with the PZT DWs. During transfer, we aligned the a -axis (zigzag orientation) of MoS₂ with the horizontal ([100]) DWs of PZT. After transfer, we used SHG mapping to check if we have achieved the desired alignment (Fig. R8g-h). During transfer, the uncertainty angle of alignment is less than 1°. There is, however, additional angular uncertainty during domain writing on PZT due to the thermal drift of the

AFM tip (up to 5°). So the overall uncertainty in the stacking angle is 2° - 6° . We included this discussion in Supplementary Section 5, and added the following sentences in the first paragraph on page 16:

“The uncertainty of stacking angle θ is 2° - 6° . The details of the sample alignment during transfer can be found in the Supplementary Information (Section 2 and Fig. S4).”

Based on the estimated uncertainty, we updated Supplemental Table S2 as shown below. Overall, the simulated intensity distribution captures well the main features of the SHG tailoring pattern in the experimental results (Fig. 3e). Large deviation from the simulated values only occurs when the DW is in the vicinity of a cracked area in MoS_2 , which compromises the net SHG signal.

Top-DW			Bottom-DW			Left-DW			Right-DW		
θ	Theory	Exp.	θ	Theory	Exp.	θ	Theory	Exp.	θ	Theory	Exp.
$0^\circ \pm 2^\circ$	4.41 ± 0.00	4.1 ± 0.3	$0^\circ \pm 2^\circ$	0.02 ± 0.01	0.0 ± 0.1	$0^\circ \pm 2^\circ$	2.00 ± 0.01	2.1 ± 0.2	$0^\circ \pm 2^\circ$	2.00 ± 0.01	1.9 ± 0.1
$15^\circ \pm 2^\circ$	4.24 ± 0.05	3.6 ± 0.2	$15^\circ \pm 2^\circ$	0.30 ± 0.08	0.2 ± 0.1	$15^\circ \pm 6^\circ$	2.1 ± 0.1	2.6 ± 0.2	$15^\circ \pm 5^\circ$	2.10 ± 0.04	1.4 ± 0.1
$30^\circ \pm 2^\circ$	3.77 ± 0.08	3.5 ± 0.1	$30^\circ \pm 2^\circ$	1.0 ± 0.1	0.6 ± 0.1	$30^\circ \pm 4^\circ$	2.5 ± 0.2	3.1 ± 0.2	$30^\circ \pm 5^\circ$	2.08 ± 0.08	0.8 ± 0.1
$45^\circ \pm 2^\circ$	3.11 ± 0.09	3.0 ± 0.1	$45^\circ \pm 3^\circ$	1.7 ± 0.1	0.7 ± 0.1	$45^\circ \pm 6^\circ$	3.1 ± 0.3	3.3 ± 0.2	$45^\circ \pm 6^\circ$	1.7 ± 0.2	0.6 ± 0.2

Table S2 | Theory-experiment comparisons of reflected SHG amplitude at 1L MoS_2 /PZT DW.

The experimental values are normalized to the average intensity difference between the Bottom-DW (set as 0) and vertical DWs (set as 2) at stacking angle $\theta = 0^\circ$.

Comment 3.7:

e) Arbitrary units are used throughout for the SHG maps. Please provide a common reference, or absolute values, allowing to compare the intensity levels across different measurements, in particular those of Fig. 2a-h, Fig. 4a-c, Fig. S3a-h.

Reply:

The SHG mapping plots shown in the paper are the raw data collected by PMTs without modification using the same experimental setup, so the intensity levels can be directly compared if they were taken at the same excitation laser power. In the revised Fig. 4f (Fig. R7d), we compared the absolute SHG intensity on the P_{up} and P_{down} domains for both reflection and transmission modes.

As the PMTs were set on photon count mode during measurement, the SHG response is not calibrated to its corresponding light intensities in the units of W/m^2 . We expressed the measured SHG strength in terms of arbitrary units (a.u.), which is proportional to the actual signal intensity measured by the PMTs. To clarify this point, we specified the excitation laser power either on the SHG image or in the figure caption, and added following sentence to the first paragraph on Page 18:

“The SHG mapping plots, unless otherwise specified, are the raw data collected by the PMTs without modification. During the SHG measurements, the PMTs in the LSM were set on the photon count mode, so the responses of the PMTs are proportional to the number of actual SHG photons detected by the PMTs, but are not calibrated to the light intensity in units of W/m^2 . The SHG mapping results

were expressed in terms of arbitrary units (a.u.), and were proportional to the actual SHG intensity detected by the PMTs. The intensity level can be directly compared if they were taken at the same excitation laser power.”

Below are the comments in the attached pdf file.

Comment 3.8:

Page 3 -lines 19-20. When introducing the PZT material system, its crystallographic axes and the experimental reference frame should also be defined (this is done at some point in the Suppl. Information, but it is good to have this information at hand for all data. It would also eliminate possible ambiguities in using the terms ‘parallel’ and ‘horizontal’, ‘vertical’ in the text, see below)

Reply:

Following the reviewer’s advice, we made the following revisions to the main text:

1. We defined the experimental reference frame and the crystalline orientation of PZT in Fig. 1 as insets, and labelled them wherever it fits.
2. We modified the following sentences in the last paragraph on Page 3:

“The PZT films are (001)-oriented with out-of-plane polar axis (Supplementary Fig. S1). Selected 1L MoS₂ flakes were transferred on top of the PZT thin film above a region patterned with a series of square domains with alternating up (P_{up} or [001]) and down (P_{down} or [00 $\bar{1}$]) polarization. ... where the horizontal (vertical) DWs are along the [100] ([010]) orientation of PZT.”

3. We specified the crystalline orientation after the terms “horizontal” and “vertical” throughout the text. For example, we revised the sentences in the 1st paragraph on Page 4 as follows:

“During transfer, the a -axis of MoS₂ (zigzag orientation) was aligned with the horizontal DWs ([100] orientation of PZT) (see Methods for transfer details).”

Comment 3.9:

Page 4 – lines 12-14 and Suppl Information Fig. S5c. As discussed in the comments for authors: further measurement data and related information should be provided concerning SHG on MoS₂-alone, MoS₂ on +z PZT and MoS₂ on -z PZT, both in transmission and reflection SHG, as references. If measurements on stand-alone MoS₂ films are not available, the role of the substrate and its polarity in both transmission and reflection measurements should be investigated and commented upon. Please note also that the text refers to MoS₂ flakes on gel films, while the title of Fig. S5c mentions MoS₂ on PZT.

Reply:

We have compared the reflected and transmitted SHG responses of MoS₂ on Gel-Film and PZT P_{up} and P_{down} domains, as discussed in detail in the reply to Comment 3.5. Measurements on Gel-Films were taken at a significantly lower excitation power to avoid damage to the soft substrate (Figs. R7c-d). On Gel-Films, the SHG signal is similar in both detection modes. While measurements on standalone MoS₂ is not available, we do not expect there’s any difference between the reflected and transmitted SHG.

To highlight the role of the substrate, we revised the following sentences in the last paragraph on Page 13:

“While bare PZT domains (Supplementary Fig. S3) and MoS₂ on Gel-Films (Supplementary Fig. 4) exhibit similar SHG responses in the reflection and transmission modes, the MoS₂/PZT heterostructure reveals qualitatively different SHG tailoring effect in these two detection modes.”

We also fixed the typo mentioned by the reviewer.

Comment 3.10:

Page 4, lines 14-15. “As the incident light is along the polar axis...”, please reformulate better, e.g.: “As the incident light is A TRANSVERSELY-POLARIZED (x,y) ELECTROMAGNETIC WAVE PROPAGATING along the polar axis...” (if this guess is correct)

Reply:

We revised the first paragraph on Page 5 according to the reviewer’s suggestion:

“For the PZT films, as the incident light is a transversely polarized (within x - y plane) electromagnetic wave propagating along the polar axis ($-z$ -direction or $[00\bar{1}]$ orientation of PZT), there is no SHG response on the uniformly polarized domains.”

We also defined the experimental reference frame, the crystalline orientations of MoS₂ and PZT, and the orientations of the polarizer and analyzer as insets in all relevant figures, as discussed in the reply to Comment 3.4.

Comment 3.11:

Page 5, lines 2-3. “the SHG signals in the transmitted mode exhibit qualitatively similar behaviors as in the reflected mode (Supplementary Fig. S3).” Please provide also scales for the signal intensities to enable a comparison (instead of a.u. maybe normalize to the same value).

Reply:

A detailed discussion can be found in the reply to Comment 3.7. We also added the following sentences in the last paragraph on Page 13 as follows:

“Overall, the maximum intensity for the transmitted light is comparable or lower than that for the reflection mode depending on PZT thickness, as the signal is collected through the oxide layers (Supplementary Fig. S4g-h).”

Comment 3.12:

Page 5, lines 14-15. Please modify the following misleading statement: “clearly showing that the interfacial SHG signal is not a simple summation of the signal strength from each constituent layer. The horizontal DW, more surprisingly, exhibit alternating ...”. In electromagnetic terms the interfacial SHG signal is equal to the sum of the fields arising from each constituent layer, as indeed seen in the experiments. Furthermore, from a SHG perspective, the results obtained on the interfacial layer are not too surprising in terms of the known symmetries of MoS₂ and Neel-like DWs in PZT

(ref. 25) and of the way in which they should compose in terms of SH polarization. Without resorting to DFT calculations, the results can be interpreted in terms of SH polarization components and related SHG tensors. This is briefly touched upon in Supplementary information but is worth a more thorough explanation and analysis as it provides the underlying theory to interpret all experimental results in Fig. 2 and 3. Specifically, for the ultra-thin layers (\ll optical wavelength) considered in the paper, basic nonlinear electromagnetic theory would predict the d -matrix (SHG tensor) of the composite system (d_{ij} interface of p. 8 in Suppl. Inf.) to be equal to the sum of the d -matrices of the adjacent layers, i.e. d -MoS₂ (Suppl Information, p. 7) and d -**DW (Suppl Information, p. 8), just because the FF field is the same and the SH polarization vectors from the MoS₂ and PZT-DW contributions do add up. This implies that the nonlinear matrix components d_{yxx} shall either add up constructively (in the case of: $d_{MoS2} + d_{Top-DW}$) or destructively (in the case of: $d_{MoS2} + d_{Bottom-DW}$) for the horizontal DWs, explaining the results in fig. 2f. With the same methodology, looking at the d_{xxx} components, one can easily justify why the MoS₂ layer does affect SHG from the vertical DWs (Fig. 2h). Related derivations and explanations should be included in the Supplementary Information and used in the text as they provide the general framework for a correct interpretation of the SHG data.

Reply:

We thank the reviewer for the valuable comments, and made the following revisions to the main text.

1. We removed this misleading sentence on Page 6 (Page 5 in previous version).
2. We added the details of the d -tensor calculation for the reflected SHG at stacking angle $\theta = 0$ (e.g., Fig. 2) to the main text on Pages 10-12, and included the full modeling details for the stacking angles-dependence of the SHG response (e.g., Fig. 3f) to the Supplementary Information Section 5. Please see the reply to Comment 3.3 for details.

Comment 3.13:

Page 5, lines 15-22 and Suppl. Information. *The alternating enhancement/suppression of the SHG signals can be easily explained with the d -tensors. Please include the reference frame in the main text, expand the analysis of Supplementary Information providing the explicit expression of the overall tensors for the (MoS₂+DW) and (MoS₂+PZT) combinations and refer to the former to clarify the interplay of the tensor symmetries and the observed SHG results (Fig. 2) in the main text. In doing so, it would also be appropriate to differentiate among the different components of the d -matrices (instead of using just one coefficient, $*A*$ or $*B*$ for each material), especially since they can be substantially different with non-negligible impact on the SHG amplitudes for different polarization configurations (e.g. in PZT the difference between d_{31} , d_{15} and d_{33} components yields differences of more than one order of magnitude in SHG intensities).*

Reply:

Following the reviewer’s suggestions, we have exploited the d -tensor element for each specific material in the SHG modeling, as detailed in the reply to Comment 3.3. To simplify the calculation, we assumed that the SHG contributions from MoS₂ and PZT DW have the same maximum intensity ($d_{MoS_2} = d_{33} \approx 0.9d_{15} = 1$), which is reasonable given their comparable dipole moments. The interfacial composite tensors can thus be reduced to:

$$d_{Top-DW}^{interface} = \begin{pmatrix} 0 & 0 & 0 & 0 & 0 & 2.1 \\ 2.1 & 0 & 1.1 & 0 & 0 & 0 \\ 0 & 0 & 0 & 1.1 & 0 & 0 \end{pmatrix},$$

$$d_{Bottom-DW}^{interface} = \begin{pmatrix} 0 & 0 & 0 & 0 & 0 & -0.1 \\ -0.1 & -2 & -1.1 & 0 & 0 & 0 \\ 0 & 0 & 0 & -1.1 & 0 & 0 \end{pmatrix},$$

$$d_{Left-DW}^{interface} = \begin{pmatrix} 1 & 1.1 & 1.1 & 0 & 0 & 1 \\ 1 & -1 & 0 & 0 & 0 & 1.1 \\ 0 & 0 & 0 & 0 & 1.1 & 0 \end{pmatrix},$$

$$d_{Right-DW}^{interface} = \begin{pmatrix} -1 & -1.1 & -1.1 & 0 & 0 & 1 \\ 1 & -1 & 0 & 0 & 0 & -1.1 \\ 0 & 0 & 0 & 0 & -1.1 & 0 \end{pmatrix},$$

$$d_{P_{up}}^{interface} = \begin{pmatrix} 0 & 0 & 0 & 0 & 1.1 & 1 \\ 1 & -1 & 0 & 1.1 & 0 & 0 \\ 1.1 & 1.1 & 1 & 0 & 0 & 0 \end{pmatrix},$$

$$d_{P_{down}}^{interface} = \begin{pmatrix} 0 & 0 & 0 & 0 & -1.1 & 1 \\ 1 & -1 & 0 & -1.1 & 0 & 0 \\ -1.1 & -1.1 & -1 & 0 & 0 & 0 \end{pmatrix}.$$

We included this discussion in the Supplementary Section 5, and updated the SHG simulation based on the revised d -tensors, which are shown in Fig. 3f and Table S2. The updated results are qualitatively similar as the previous version.

Comment 3.14:

Page 6, line 18. “parallel” in this context means probably ‘horizontal’ (DW). In any case it is better to remove this ambiguity by introducing and using a common x,y,z reference frame (1st comment).

Reply:

We have followed the reviewer’s suggestion, fixed the typo, and specified the orientation of DWs after “horizontal” and “vertical” terms. Please see the reply to Comment 3.8 for details.

Comment 3.15:

Pages 7-8. *Theoretical modelling.* As mentioned in the comments to Authors, apart from DFT simulations on the polar coupling, concerning ultimately only one direction (//) the symmetries of the problem should be introduced and analysed in the more general theoretical framework of SHG tensors (d -matrices). This applies to the expressions of the d -matrices to be used in the analysis of the data of Fig. 2 and Fig. 3 (and table S2). Hence also the explicit expression of d_{ij} interface as a function of θ should also be included in Supp. Information.

Reply:

Following the reviewer’s suggestion, we have derived the explicit expression of $d_{ij}^{interface}$ as a function of stacking angle (θ), and added them to the Supplementary Information (Section 5, Eq. S5). We also added the following sentence to the 1st paragraph on Page 12:

“Furthermore, we derived the explicit expressions of interfacial SHG tensors at the four DWs as a function of stacking angle θ , which are given in the Supplementary Information (Eq. S5).”

Comment 3.16:

Page 9-10. Transmitted SHG response. This part requires thorough revision.

- *The Methods part or Suppl. Information should include:*
 - *details on the experimental setup used the SHG transmission measurements: what was the optics for signal collection (medium underneath the PZT sample, type of objective, magnification, NA, etc.)*
 - *information of the bottom facet of the PZT sample (was PZT deposited on another substrate? If so which one & how thick? Which other interfaces are traversed by the signals in transmission mode?)*
- *The focusing conditions (in particular the spot size) of the fundamental beam should be specified in the methods section (both for transmission and reflection measurements)*

Reply:

We added detailed information of the experimental setup for SHG transmission measurement and the focusing conditions of the fundamental beam to the Method section. Please see the reply to Comment 3.4.

In our work, we deposited 20-50 nm thick epitaxial PZT on $\text{La}_{0.67}\text{Sr}_{0.33}\text{MnO}_3$ (LSMO) (10 nm) buffered (001) SrTiO_3 substrates (0.5 mm) via off-axis radio frequency magnetron sputtering. We added the information to the Supplementary Section 1 and Method section (2nd paragraph on Page 15) in the main text as follows:

“We deposited 20-50 nm thick epitaxial PZT films on 10 nm $\text{La}_{0.67}\text{Sr}_{0.33}\text{MnO}_3$ (LSMO) buffered (001) SrTiO_3 substrates (5 mm×5 mm×0.5 mm) via off-axis radio frequency magnetron sputtering.”

As discussed in the reply to Comment 3.2, considering the wavelength of the SHG signal, the relevant dielectrics for focusing in the transmission mode are the PZT/LSMO/STO stack.

Comment 3.17:

- *The claims and justifications for the coupling between “the out-of-plane bulk polarization of PZT” and “the polar axis of MoS₂” need to be substantiated with other arguments and further clarifications.*
- *The notation used in Fig. 4 is misleading, whereby the propagation directions of the fields (normally labelled by the wavevector k) are labelled as electric (E -) fields, which – in the case of ordinary TEM waves – are polarized orthogonally to the former (k).*
- *Incorrect statements made on page 10 (highlighted in the comments for authors) should also be removed.*

Reply:

Please see the reply to Comment 3.2.

With the new data and revisions we provided, we hope that we have satisfactorily addressed all the comments of the reviewers', and our manuscript is now suitable for publication in *Nature Communications*.

List of Changes:

Changes to the author list

1. We added a new co-author, Yifei Hao, who prepared 20 nm and 30 nm thick PZT films. We also revised the following sentence in the Author Contributions: "L.Z. and Y.H. prepared the PZT thin films."

Changes to the figures

2. We defined the experimental reference frame, the crystalline orientations of MoS₂ and PZT, and the orientations of the polarizer and analyzer as insets in all relevant figures.
3. We updated the simulated SHG results in Fig. 3f based on the revised *d*-tensor calculation.
4. We replaced the original Fig. 4f in the main text. The new Fig. 4f summarizes the SHG intensity on P_{up} and P_{down} domains in both reflection and transmission modes as a function of PZT thickness.

Changes to the main text

We made the following revisions to the main text.

5. Abstract, last sentence: "Unlike the extensively studied coupling mechanisms driven by charge, spin, and lattice, the interfacial tailoring effect is solely mediated by the polar symmetry of the constituent layers..."
6. We specified the crystalline orientations of the ferroelectric domain walls after the terms "horizontal" and "vertical" throughout the text.
7. Page 3, last paragraph: "For the ferroelectric layer, we worked with 20-50 nm thick epitaxial PZT thin films deposited on (001) SrTiO₃ substrates, with La_{0.67}Sr_{0.33}MnO₃ (LSMO) (10 nm) buffer layers serving as the bottom electrode (Methods). The PZT films are (001)-oriented with out-of-plane polar axis (Supplementary Fig. S1). Selected 1L MoS₂ flakes were transferred on top of the PZT thin film above a region patterned with a series of square domains with alternating up (P_{up} or [001]) and down (P_{down} or [00 $\bar{1}$]) polarization. ... where the horizontal (vertical) DWs are along the [100] ([010]) orientation of PZT."
8. Page 4, 1st paragraph: "During transfer, the *a*-axis of MoS₂ (zigzag orientation) was aligned with the horizontal DWs ([100] orientation of PZT) (see Methods for transfer details). As shown in Fig. 1d, the presence of the MoS₂ top layer does not alter the underneath domain structure. This is not surprising as the PZT film is exposed to the ambient condition prior to the transfer, where the polarized surface bound charge can be well screened by charged adsorbates.^{26, 27} The MoS₂ flake is deposited on top of the domain structure via a dry-transfer approach, which should not affect this surface screening layer on PZT."
9. Page 4, 2nd paragraph: "Such modulation of PL spectra in TMDCs via neighboring ferroelectric domains has previously been attributed to the polarization induced doping effect,^{23, 24} and confirms

the close interfacial contact between MoS₂ and PZT in our samples. The relative strength of the modulation, however, can be affected by the surface screening layer on PZT,²⁷ and thus depends on the preparation details of the composite structures (Supplementary Section 2).^{26,}

10. Page 4, last paragraph: “For normal incident light (800 nm center wavelength), we observed strong SHG response (~400 nm) from the 1L MoS₂ flakes on Gel-Films, which conforms to the rotational symmetry of the lattice (Supplementary Figs. S4b-d). For the PZT films, as the incident light is a transversely polarized (within x - y plane) electromagnetic wave propagating along the polar axis ($-z$ -direction or $[00\bar{1}]$ orientation of PZT), there is no SHG response on the uniformly polarized domains.”
11. Page 5, 1st paragraph: “The width of the detected SHG signal is about 300-400 nm, which approaches the diffraction limit at this wavelength and the resolution of the SHG microscope (Methods). ...For both MoS₂ on Gel-Films and bare PZT, the SHG signals detected in the transmission mode exhibit qualitatively similar behavior as in the reflection mode (Supplementary Fig. S3 and Fig. S4c-d).”
12. Page 5, last paragraph: “Unlike the PL data (Figs. 1e-f), no prominent difference in the SHG signal has been observed in the regions on the P_{up} and P_{down} domains, confirming the signal is not affected by the interfacial charge coupling between MoS₂ and PZT.”
13. Page 6, 1st paragraph: “The tailoring of reflected SHG signal at the DW is a robust effect and has been observed in multiple 1L MoS₂ samples. Similar tuning pattern is also observed on three-layer and five-layer MoS₂ flakes on PZT, and is absent in bilayer and four-layer MoS₂ (Supplementary Fig. S6). This observation clearly demonstrates that the observed effect originates from the noncentrosymmetric symmetry of MoS₂. The intensity of the SHG signal gradually diminishes in thicker MoS₂, confirming that it is an interface rather than bulk phenomenon.”
14. Page 7, 2nd paragraph: “To conform to the bulk polarization change, the surface polarization at the vicinities of the odd and even DWs is expected to have opposite chirality (Fig. 3a), with the corresponding \vec{p}_{\parallel} pointing to $-y$ and $+y$ directions, respectively.”
15. Page 7, last paragraph: “We further compared the SHG response of 1L MoS₂ interfaced with 20 nm, 30 nm, and 50 nm PZT films (Supplementary Fig. S7). Despite the different PZT thicknesses, all heterostructures exhibit qualitatively similar SHG responses, with alternating enhancement and suppression of the SHG signal observed at the horizontal ($[100]$) DWs and unappreciable SHG contrast observed at the vertical ($[010]$) DWs. This result further confirms the interfacial nature of the DW’s tailoring effect. In fact, the 180° DW in bulk PZT is on the order of a couple of unit cells and does not acquire an in-plane component.^{33, 34} The chiral rotation of the local dipole, which is critical for forming the in-plane polarization, can only be stabilized at the surfaces/interfaces (Fig. 3a) due to the presence of strong depolarization field.^{35, 36} For example, previous transmission electron microscopy (TEM) studies have revealed a flux-closure polar structure at the surface of the DW in PZT thin films,³⁵ and even emergence of polar vortices in PbTiO₃/SrTiO₃ superlattices,^{6, 7} where theoretical modeling has pointed to the dominant role of the interface contribution to the polar anomaly.⁷”
16. Page 8, 2nd paragraph: “To examine the feasibility of the interfacial polar coupling scenario, we exploited a phenomenological model to estimate the net in-plane polarization at the DW based on the TEM study in Ref. [35], considering a triangle-shaped flux-closure domain structure that hosts continuous electric dipole rotations at the surface of a 180° DW (Fig. 3a). ...Compared with an a -domain like DW configuration induced by the local electric field of a biased AFM tip (Supplementary

Fig. S10), this model depicts a dipole distribution with comparable (if not smaller) spatial extension but lower electrostatic energy.³⁴”

17. Page 9, 2nd paragraph: “While the precise value of the polarization may vary, this simple model naturally explains the major features of our observation. When one of the polar axes of MoS₂ is aligned with the in-plane polarization of PZT at the DW regions, as for the odd DWs, their excited interfacial SH dipole fields are coherently coupled,^{37, 38} leading to significantly enhanced SHG response that is linearly polarized along the polar axis.”
18. We added a detailed discussion of the *d*-tensor modeling of the reflected SHG response on pages 10-12.
19. Page 12, last paragraph: “Comparing the results shown in Fig. 2b and Fig. 2e, it is clear that the lateral polarization of PZT DW can replace an optical analyzer to provide efficient filtering of the light polarization for the SHG signal of MoS₂. It further enhances or quenches the SHG intensity for the selected light polarization depending on the underlying DW chirality. Compared with the existing optical filter technologies, which are macroscopic in terms of dimensions, time consuming in terms of optical setup, and cannot be programmed at the nanoscale, the MoS₂/PZT heterostructure has the distinct advantages in terms of size scaling and being nanoscale reconfigurable, offering the opportunities to achieve on-chip generation and smart filtering of SHG signals for nano-optics.”
20. Page 13, 2nd paragraph: “While bare PZT domains (Supplementary Fig. S3) and MoS₂ on Gel-Films (Supplementary Fig. S4) exhibit similar SHG responses in the reflection and transmission modes, the MoS₂/PZT heterostructure reveals qualitatively different SHG tailoring effect in these two detection modes. ... Overall, the maximum intensity for the transmitted light is comparable or lower than that for the reflection mode depending on PZT thickness, as the signal is collected through the oxide layers (Supplementary Fig. S4g-h).”
21. We added a detailed discussion of the transmitted SHG results on Page 14.
22. Page 15, 1st paragraph: “The transmitted SHG signal, in sharp contrast, is sensitively tuned by the out-of-plane ferroelectric domain rather than the DW, which is attributed to the bulk state of PZT.”
23. Page 15, 2nd paragraph: “We deposited 20-50 nm thick epitaxial PZT films on 10 nm La_{0.67}Sr_{0.33}MnO₃ (LSMO) buffered (001) SrTiO₃ substrates (5 mm×5 mm×0.5 mm) via off-axis radio frequency magnetron sputtering.”
24. Page 16, 1st paragraph: “The uncertainty of stacking angle θ is 2°-6°. The details of the sample alignment during transfer can be found in the Supplementary Information (Section 2 and Fig. S4).”
25. Page 16, 2nd paragraph: “The resolution of PFM is about 5 nm for our experimental setup,⁴¹ which cannot resolve the intrinsic DW width.”
26. We added detailed information of the experimental setup for the SHG measurement and the focusing condition for the fundamental beam to the Method section (Pages 16-18).

Changes to the references

27. We added nine references in the main text:

[7] Zubko, P. et al. Negative capacitance in multidomain ferroelectric superlattices. *Nature* **534**, 524 (2016).

- [22] Lipatov, A., Li, T., Vorobeva, N.S., Sinitskii, A. & Gruverman, A. Nanodomain Engineering for Programmable Ferroelectric Devices. *Nano Letters* **19**, 3194-3198 (2019).
- [24] Wen, B. et al. Ferroelectric-Driven Exciton and Trion Modulation in Monolayer Molybdenum and Tungsten Diselenides. *ACS Nano* **13**, 5335-5343 (2019).
- [25] Lv, L. et al. Reconfigurable two-dimensional optoelectronic devices enabled by local ferroelectric polarization. *Nature Communications* **10**, 3331 (2019).
- [26] Song, J.F. et al. Enhanced Piezoelectric Response in Hybrid Lead Halide Perovskite Thin Films via Interfacing with Ferroelectric $\text{PbZr}_{0.2}\text{Ti}_{0.8}\text{O}_3$. *ACS Applied Materials & Interfaces* **10**, 19218-19225 (2018).
- [27] Hong, X. et al. Unusual resistance hysteresis in n-layer graphene field effect transistors fabricated on ferroelectric $\text{Pb}(\text{Zr}_{0.2}\text{Ti}_{0.8})\text{O}_3$. *Applied Physics Letters* **97**, 033114 (2010).
- [34] Hlinka, J. & Marton, P. Phenomenological model of a 90 degrees domain wall in BaTiO_3 -type ferroelectrics. *Physical Review B* **74**, 104104 (2006).
- [37] Kurimura, S. & Uesu, Y. Application of the second harmonic generation microscope to nondestructive observation of periodically poled ferroelectric domains in quasi-phase-matched wavelength converters. *Journal of Applied Physics* **81**, 369-375 (1997).
- [38] Kaneshiro, J., Uesu, Y. & Fukui, T. Visibility of inverted domain structures using the second harmonic generation microscope: Comparison of interference and non-interference cases. *Journal of the Optical Society of America B-Optical Physics* **27**, 888-894 (2010).

Changes to the Supplementary Information

We made the following revisions to the Supplementary Information.

28. We added the PFM phase and amplitude images of the PZT domain structure shown in Fig. 1c (Figs. S1c-d).
29. Section 2: We expanded the discussion of the transfer and alignment of MoS_2 on PZT (Fig. S4) and the polarization-induced PL modulation.
30. Section 3: We added detailed discussions of how the SHG response varies with the layer thickness of MoS_2 (Fig. S6) and PZT (Fig. S7), as well as the depth dependence of the transmitted SHG signal (Fig. S8).
31. Section 4: We added the discussion of the possible effect of a biased AFM tip on local dipole orientation within the DW (Fig. S10).
32. Section 5: We updated the SHG modeling using the second-order nonlinear optical susceptibility tensors, and included the comparison of the theoretical and experimental results of the reflected SHG responses.

Reviewers' Comments:

Reviewer #1:

Remarks to the Author:

In the previous report, the reviewer made six questions/comments (from 1.1 to 1.6) on this paper. Among these, the authors appropriately answered to 1.2 to 1.6, and they revised the paper to reflect it properly.

The question 1.1, on the other hand, concerns the interpretation of the most important experimental results (Fig. 2) in this paper: The question is whether this effect is due to (a) the interfacial effect between MoS₂ and PZT films as the authors insist or (b) due to SHG interference from the respective bulk of MoS₂ and PZT. The authors capture this question accurately and conducts new elaborate experiments. The reviewer evaluates this point. However, in order to clarify that the cause is not (b) but (a), the following must be fully considered: That is, in the case of (a), the combined SH amplitude Ψ_{total} does not depend on the thickness of MoS₂, but in the case of (b), Ψ_{total} depends on it through the following equation.

$$\psi_{\text{total}} = \psi_{\text{PLZ}} + \psi_{\text{MoS}_2} e^{-i\phi}$$

where ϕ is a relative phase and is expressed using the wavelength λ of the fundamental wave, the refractive index n and the thickness d of a MoS₂ film, as

$$\phi = (2\pi/\lambda)n \cdot d$$

With $\lambda = 800$ nm, $d = 0.311$ nm (p9, line 8) and an assumed value of n (~ 1.5), we get $\phi = 0.42$ degrees of arc for 1L MoS₂ film. That is, in order to prove that it is not (a), it is necessary to change the phase by up to 180 °, but for that purpose, the MoS₂ thickness must be at least two orders of magnitude thicker than the one used. The reviewer would like to hear the author's thoughts on this point.

On the other hand, since the domain boundary having the chirality of the polarization (Fig.3a) is limited to the top surface of the PZT thin film, and does not depend on the thickness of PZT. It has been proven by the authors' experiments.

Reviewer #2:

Remarks to the Author:

The authors have substantially improved the manuscript and answered most questions of the reviewers.

There is still the question about the "interfacial polar symmetry", which needs further explanation. The authors argued that the D_{3h} symmetry of MoS₂ is reduced to C_{3v} by the coupling to vertical polarization of PZT. Here the D_{3h} symmetry is only for MoS₂, while the C_{3v} is for MoS₂ + PZT with vertical polarization. There is no symmetry coupling, but just simply expand the atomic set to include both sets. In other words, the "unit cell" is enlarged so that the symmetry of the new larger set becomes lower. The symmetry of solid material structures can be changed by either changing the atomic arrangement, like phase transitions, or by adding new atoms to form a larger "unit cell", like the case discussed here. The authors used the language of "coupling two symmetries to form a new symmetry" is very confusing. In general, we do not need to create fancy words to make it more eye catching but confusing if the fact can be simply describes by readily available and understandable language. If I am not mistaken, the straightforward statement of what the authors have done is as follows: The generated SHG signal of one component is filtered by another component of a thin film+substrate system and the filtering characteristics are determined by the symmetries of both components. There is no "interfacial polar symmetry"!

If the main purpose of this work is to introduce a new filtering mechanism, what people care about is the efficiency of this method compared to available filtering methods, like using an analyzer, the uniqueness in certain applications, easy fabrication, cost, etc. As I said before, the phenomenon is very interesting and I would like to see it can be liked to certain potential applications to demonstrate the value of this finding.

We thank the reviewers for reviewing our manuscript and appreciate the reviewers' valuable comments/suggestions. We have revised the manuscript and Supplementary Information based on the reviewers' advices. A complete list of changes can be found at the end of the response letter.

Response to Reviewer #1's Comments

Comment:

In the previous report, the reviewer made six questions/comments (from 1.1 to 1.6) on this paper. Among these, the authors appropriately answered to 1.2 to 1.6, and they revised the paper to reflect it properly.

The question 1.1, on the other hand, concerns the interpretation of the most important experimental results (Fig. 2) in this paper: The question is whether this effect is due to (a) the interfacial effect between MoS₂ and PZT films as the authors insist or (b) due to SHG interference from the respective bulk of MoS₂ and PZT. The authors capture this question accurately and conducts new elaborate experiments. The reviewer evaluates this point. However, in order to clarify that the cause is not (b) but (a), the following must be fully considered: That is, in the case of (a), the combined SH amplitude Ψ_{total} does not depend on the thickness of MoS₂, but in the case of (b), Ψ_{total} depends on it through the following equation.

$$\psi_{total} = \psi_{PLZ} + \psi_{MoS_2} e^{-i\phi}$$

where ϕ is a relative phase and is expressed using the wavelength λ of the fundamental wave, the refractive index n and the thickness d of a MoS₂ film, as

$$\phi = (2\pi/\lambda)n \cdot d$$

With $\lambda = 800$ nm, $d = 0.311$ nm (p9, line 8) and an assumed value of n (~1.5), we get $\phi = 0.42$ degrees of arc for 1L MoS₂ film. That is, in order to prove that it is not (a), it is necessary to change the phase by up to 180 °, but for that purpose, the MoS₂ thickness must be at least two orders of magnitude thicker than the one used. The reviewer would like to hear the author's thoughts on this point.

On the other hand, since the domain boundary having the chirality of the polarization (Fig.3a) is limited to the top surface of the PZT thin film, and does not depend on the thickness of PZT. It has been proven by the authors' experiments.

Reply:

After carefully considering the reviewer's comments, we find that our proposed interfacial coupling model does not contradict the concept of interference. As discussed in the last paragraph on Page 9 in the manuscript, we implicitly assumed that the SH dipole fields of two constituent materials are coherently coupled when we modeled the composite second-order optical susceptibility tensor for the MoS₂/PZT DW heterointerface, $\chi_{interface}^{(2)} = \chi_{MoS_2}^{(2)} + \chi_{DW}^{(2)}$. This assumption is valid because the phase change due to the light propagation in monolayer MoS₂ is negligibly small, as estimated by the reviewer. Given this assumption, mathematically, scenarios (a) and (b) would yield essentially the same result, as the reviewer also noted.

On the other hand, we believe that the interfacial polar coupling model better highlights two central features of our observation: 1) the filtering of the SHG polarization depends critically on the polar symmetry and the alignment of the polar axes of these two constituent materials; and 2) the synergetic behavior remains to be within nanometer scale distance from the interface.

The first point is clearly illustrated in the MoS₂ layer-dependence of the SHG tailoring effect (SI Fig. S6d), where only odd-layer (1L, 3L, and 5L) MoS₂ can yield a tailored SHG response at the DW. The effect is absent in even-layer (2L and 4L) MoS₂ as they are centrosymmetric and thus do not generate net SHG signal. While in principle this effect can be interpreted within the interference model, it can be directly deduced from the polar symmetry of the interfacial constituent layers.

The second point lies in the fact that pronounced SHG response can only be observed in few-layer MoS₂ samples. In Fig. S6d, it is clear that the SHG intensity decreases rapidly with increasing MoS₂ layer numbers and eventually diminishes in regions thicker than 10 layers. This effect was previously reported in literature (Ref. 29), which shows that the SHG signal from odd-layer MoS₂ decreases significantly from 1L to 5L (Fig. R1). Such rapid attenuation cannot be accounted for simply by the propagation phase shift and absorption, and has been attributed to the interlayer coupling induced modification of the linear and nonlinear susceptibility. In the bulk limit, the SH response is predicted to be only 0.2% of the 1L MoS₂ (Ref. 29).

[Redacted]

Fig. R1. Layer number dependence of SH intensity (histogram) for few-layer MoS₂. The red squares are the modeling result that only considers the layer phase shift and absorption of the fundamental and SH fields. Adapted from Ref. 29: Li *et al.*, *Nano Lett.* 13, 3329 (2013).

In summary, we believe that our proposed interfacial polar coupling model does not contradict the interference model proposed by the reviewer, while it better captures the interfacial nature and synergetic effect of the SHG response for MoS₂ and PZT DW. To clarify this point, we revised the first paragraph on Page 6 in the main text as follows:

“In samples with odd-layer MoS₂, the modulation strength decreases with increasing layer number, consistent with fact that the SHG signal of MoS₂ attenuates rapidly in thicker films.²⁹”

We also revised the following sentence in the 5th paragraph on Page 7 in the Supplementary Information.

“The gradually diminishing tuning contrast in thicker MoS₂ agrees with fact that the SHG signal of MoS₂ attenuates rapidly with increasing layer numbers, confirming that it is an interfacial rather than

bulk phenomenon (Li *et al.*, *Nano Lett.*). The tailoring effect is also consistent with the coherent interference between the SH fields of the two constituent layers.”

Response to Reviewer #2’s Comments

Comment:

The authors have substantially improved the manuscript and answered most questions of the reviewers.

There is still the question about the “interfacial polar symmetry”, which needs further explanation. The authors argued that the D_{3h} symmetry of MoS₂ is reduced to C_{3v} by the coupling to vertical polarization of PZT. Here the D_{3h} symmetry is only for MoS₂, while the C_{3v} is for MoS₂ + PZT with vertical polarization. There is no symmetry coupling, but just simply expand the atomic set to include both sets. In other words, the “unit cell” is enlarged so that the symmetry of the new larger set becomes lower. The symmetry of solid material structures can be changed by either changing the atomic arrangement, like phase transitions, or by adding new atoms to form a larger “unit cell”, like the case discussed here. The authors used the language of “coupling two symmetries to form a new symmetry” is very confusing. In general, we do not need to create fancy words to make it more eye catching but confusing if the fact can be simply described by readily available and understandable language. If I am not mistaken, the straightforward statement of what the authors have done is as follows: The generated SHG signal of one component is filtered by another component of a thin film+substrate system and the filtering characteristics are determined by the symmetries of both components. There is no “interfacial polar symmetry”!

If the main purpose of this work is to introduce a new filtering mechanism, what people care about is the efficiency of this method compared to available filtering methods, like using an analyzer, the uniqueness in certain applications, easy fabrication, cost, etc. As I said before, the phenomenon is very interesting and I would like to see it can be liked to certain potential applications to demonstrate the value of this finding.

Reply:

We have carefully considered the reviewer’s comments, and agree that the reviewer’s descriptions of “an enlarged interfacial unit cell with a lower symmetry” and “*the filtering characteristics are determined by the symmetries of both components*” well capture the central observations in this work. Following the reviewer’s suggestions, we removed all phrases related to “symmetry coupling” and “interfacial polar symmetry” from the manuscript and revised the main text as follows:

Abstract: “we report an unconventional nonlinear optical filtering effect resulting from the **interfacial polar alignment** between monolayer MoS₂ and a neighboring ferroelectric oxide thin film.”

Page 3, 1st paragraph: “In this work, we report **an unconventional nonlinear optical filtering effect enabled by the polar symmetry** of 1L MoS₂ and a neighboring ferroelectric PbZr_{0.2}Ti_{0.8}O₃ (PZT) thin film.”

Page 15, 2nd paragraph: “The tailoring effect for the reflected SHG signal is solely determined by the **polar symmetry of MoS₂ and PZT DW.**”

With these revisions, we believe that we have satisfactorily addressed all the comments of the reviewers, and hope the manuscript is now suitable for publication in *Nature Communications*.

List of Changes:

1. Abstract: “we report an unconventional nonlinear optical filtering effect resulting from the **interfacial polar alignment** between monolayer MoS₂ and a neighboring ferroelectric oxide thin film.”
2. Page 3, 1st paragraph: “In this work, we report **an unconventional nonlinear optical filtering effect enabled by the polar symmetry** of 1L MoS₂ and a neighboring ferroelectric PbZr_{0.2}Ti_{0.8}O₃ (PZT) thin film.”
3. Page 6, 1st paragraph: “In samples with odd-layer MoS₂, the modulation strength decreases with increasing layer number, consistent with fact that the SHG signal of MoS₂ attenuates rapidly in thicker films.²⁹”
4. Page 15, 2nd paragraph: “The tailoring effect for the reflected SHG signal is solely determined by the **polar symmetry of MoS₂ and PZT DW**.”
5. Page 18, last paragraph: we revised the Data Availability statement as “**All relevant data that support the findings of this study are available from the corresponding authors upon request.**”
6. Supplementary Information, Page 7, 5th paragraph: “The **gradually diminishing tuning contrast in thicker MoS₂ agrees with fact that the SHG signal of MoS₂ attenuates rapidly with increasing layer numbers, confirming that it is an interfacial rather than bulk phenomenon.**² The tailoring effect is also consistent with the coherent interference between the SH fields of the two constituent layers.”

Reviewers' Comments:

Reviewer #1:

Remarks to the Author:

I read carefully the responses of the authors to the two referees. Now I think I understand what the authors intended to insist and am able to recommend the paper as an article of Nature Communications.

I've enjoyed constructive discussions with authors.

Reviewer #2:

None